# Logarithmic Smoothing for Pessimistic Off-Policy Evaluation, Selection and Learning

**Otmane Sakhi**
Criteo AI Lab, Paris, France
o.sakhi@criteo.com

**Imad Aouali**
CREST, ENSAE
Criteo AI Lab, Paris, France
i.aouali@criteo.com

**Pierre Alquier**
ESSEC Business School, Singapore
alquier@essec.edu

**Nicolas Chopin**
CREST, ENSAE
nicolas.chopin@ensae.fr

## Abstract

This work investigates the offline formulation of the contextual bandit problem, where the goal is to leverage past interactions collected under a behavior policy to evaluate, select, and learn new, potentially better-performing, policies. Motivated by critical applications, we move beyond point estimators. Instead, we adopt the principle of *pessimism* where we construct upper bounds that assess a policy's worst-case performance, enabling us to confidently select and learn improved policies. Precisely, we introduce novel, fully empirical concentration bounds for a broad class of importance weighting risk estimators. These bounds are general enough to cover most existing estimators and pave the way for the development of new ones. In particular, our pursuit of the tightest bound within this class motivates a novel estimator (LS), that *logarithmically smooths* large importance weights. The bound for LS is provably tighter than its competitors, and naturally results in improved policy selection and learning strategies. Extensive policy evaluation, selection, and learning experiments highlight the versatility and favorable performance of LS.

## 1 Introduction

In decision-making under uncertainty, offline contextual bandit [16] presents a practical framework for leveraging past interactions with an environment to optimize future decisions. This comes into play when we possess logged data summarizing an agent's past interactions [10]. These interactions, typically captured as context-action-reward tuples, hold valuable insights into the underlying dynamics of the environment. Each tuple represents a single round of interaction, where the agent observes a context (including relevant features), takes an action according to its current policy, often called *behavior policy*, and receives a reward that depends on both the observed context and the taken action. This framework is prevalent in interactive systems like online advertising, music streaming, and video recommendation. In online advertising, for instance, the user's profile is the context, the recommended product is the action, and the click-through rate (CTR) is the expected reward. By learning from past interactions, the recommender system tailors product suggestions to individual preferences, maximizing engagement and ultimately, business success.

To optimize future decisions without requiring real-time deployments, this framework presents us with three tasks: off-policy evaluation (OPE) [16], off-policy selection (OPS) [32], and off-policy learning (OPL) [55]. OPE estimates the risk: the *negative of expected reward* that a *target policy* would achieve, essentially predicting its performance if deployed. OPS selects the best-performing

38th Conference on Neural Information Processing Systems (NeurIPS 2024).

policy from a finite set of options, and OPL finds the optimal policy within an infinite class of policies. In general, OPE is an intermediary step for OPS and OPL since its primary goal is policy comparison.

A significant amount of research in OPE has centered around Inverse Propensity Scoring (IPS) estimators [24, 16–18, 60, 19, 54, 38, 32, 45]. These estimators rely on importance weighting to address the discrepancy between the target and behavior policies. While unbiased under some conditions, IPS induces high variance. To mitigate this, regularization techniques have been proposed for IPS [10, 38, 54, 5, 21] trading some bias for reduced variance. However, these estimators can still deviate from the true risk, undermining their reliability for decision-making, especially in critical applications. In such scenarios, practitioners need estimates that cover the true risk with high confidence. To address this, several approaches focused on constructing either asymptotic [10, 48, 15] or finite sample [32, 21], high probability, empirical upper bounds on the risk. These bounds evaluate the performance of a policy in the worst-case scenario, adopting the principle of pessimism [27].

If this principle is used in OPE, it is central in OPS and OPL, where strategies are inspired by, or directly derived from, upper bounds on the risk [55, 35, 32, 49, 5, 59, 21]. Examples for OPS include Kuzborskij et al. [32] who employed an Efron-Stein bound for self-normalized IPS, or Gabbianelli et al. [21] that based their analysis on an upper bound constructed with the Implicit Exploration estimator. Focusing on OPL, Swaminathan and Joachims [55] exploited the empirical Bernstein bound [36] alongside the Clipping estimator to motivate sample variance penalization. This work was recently improved by either modifying the penalization [59] or analyzing the problem from the PAC-Bayesian lens [35]. The latter direction was further explored by Sakhi et al. [49], Aouali et al. [5, 7], Gabbianelli et al. [21] resulting in tight PAC-Bayesian bounds that can be directly optimized.

Existing *pessimistic* OPE, OPS, and OPL approaches involve analyzing the concentration properties of a *pre-defined risk estimator*, often chosen to simplify the analysis. We propose a different approach: we derive general concentration bounds applicable to a broad class of regularized IPS estimators and then identify the estimator within this class that achieves the tightest concentration bound. This leads to a tailored estimator, named Logarithmic Smoothing (LS). LS enjoys several desirable properties. It concentrates at a sub-Gaussian rate, and has a finite variance without being necessarily bounded. Its concentration upper bound allows us to evaluate the worst-case risk of any policy, enables us to derive a simple OPS strategy that directly minimizes our estimator akin to Gabbianelli et al. [21], and achieves state-of-the-art learning guarantees for OPL when analyzed within the PAC-Bayesian framework akin to [35, 49, 5, 7, 21].

This paper is structured as follows. Section 2 introduces the necessary background. In Section 3, we provide unified risk bounds for a broad class of regularized IPS estimators, for which LS enjoys the tightest upper bound. In Section 4, we analyze LS for OPS and OPL, and we further extend the analysis within the PAC-Bayesian framework. Extensive experiments in Section 5 highlight the favorable performance of LS, and Section 6 provides concluding remarks.

## 2 Setting and background

**Offline contextual bandit.** Let $\mathcal{X} \subset \mathbb{R}^d$ be the *context space*, which is a compact subset of $\mathbb{R}^d$, and let $\mathcal{A} = [K]$ be a finite *action set*. An agent's actions are guided by a *stochastic* and *stationary* policy $\pi \in \Pi$ within a policy space $\Pi$. Given a context $x \in \mathcal{X}$, $\pi(\cdot|x)$ is a probability distribution over the action set $\mathcal{A}$; $\pi(a|x)$ is the probability that the agent selects action $a$ in context $x$. Then, an agent interacts with a contextual bandit over $n$ rounds. In round $i \in [n]$, the agent observes a context $x_i \sim \nu$ where $\nu$ is a distribution with support $\mathcal{X}$. After this, the agent selects an action $a_i \sim \pi_0(\cdot|x_i)$, where $\pi_0$ is the *behavior policy* of the agent. Finally, the agent receives a stochastic cost $c_i \in [-1, 0]$ that depends on the observed context $x_i$ and the taken action $a_i$. This cost $c_i$ is sampled from a cost distribution $p(\cdot|x_i, a_i)$. This leads to $n$-sized logged data, $\mathcal{D}_n = (x_i, a_i, c_i)_{i \in [n]}$, where tuples $(x_i, a_i, c_i)$ for $i \in [n]$ are i.i.d. The expected cost of taking action $a$ in context $x$ is $c(x, a) = \mathbb{E}_{c \sim p(\cdot|x,a)}[c]$, and the costs are negative because they are interpreted as the negative of rewards. The performance of a policy $\pi \in \Pi$ is evaluated through its *risk*, which aggregates the expected costs $c(x, a)$ over all possible contexts $x \in \mathcal{X}$ and taken actions $a \in \mathcal{A}$ by policy $\pi$, such as

$$R(\pi) = \mathbb{E}_{x \sim \nu, a \sim \pi(\cdot|x), c \sim p(\cdot|x,a)}[c] = \mathbb{E}_{x \sim \nu, a \sim \pi(\cdot|x)}[c(x, a)] . \tag{1}$$

The main goal is to use logged dataset $\mathcal{D}_n$ to enhance future decision-making without necessitating live deployments. This often entails three tasks: OPE, OPS, and OPL. First, OPE is concerned

with constructing an estimator $\hat{R}_n(\pi)$ of the risk $R(\pi)$ of a fixed *target policy* $\pi$ and study its deviation, aspiring for $\hat{R}_n(\pi)$ to concentrate well around $R(\pi)$. Second, OPS focuses on selecting the best performing policy $\hat{\pi}_n^{\mathrm{S}}$ from a *predefined* and *finite* collection of target policies $\{\pi_1, \ldots, \pi_m\}$, effectively seeking to determine $\operatorname{argmin}_{k \in [m]} R(\pi_k)$. Third, OPL aims to find a policy $\hat{\pi}_n^{\mathrm{L}}$ within the *potentially infinite policy space* $\Pi$ that achieves the lowest risk, essentially aiming to find $\operatorname{argmin}_{\pi \in \Pi} R(\pi)$. In general, both OPS and OPL rely on OPE's initial estimation of the risk.

**Regularized IPS.** Our work focuses on the inverse propensity scoring (IPS) estimator [24]. IPS approximates the risk of a policy $\pi$, $R(\pi)$, by adjusting the contribution of each sample in logged data according to its *importance weight (IW)*, which is the ratio of the probability of an action under the target policy $\pi$ to its probability under the behavior policy $\pi_0$,

$$\hat{R}_n(\pi) = \frac{1}{n} \sum_{i=1}^{n} w_\pi(x_i, a_i) c_i \, , \tag{2}$$

where for any $(x, a) \in \mathcal{X} \times \mathcal{A}$, $w_\pi(x, a) = \pi(a|x)/\pi_0(a|x)$ are the IWs. IPS is unbiased under the coverage assumption (see for example Owen [39, Chapter 9]). However, it can suffer high variance, which tends to scale linearly with IWs [57]. This issue becomes pronounced when there is a significant discrepancy between the target policy $\pi$ and the behavior policy $\pi_0$. To mitigate this, a common strategy consists in applying a regularization function $h : [0,1]^2 \times [-1, 0] \to (-\infty, 0]$ to $\pi(a|x)$, $\pi_0(a|x)$ and $c$. This function is designed to reduce the estimator's variance at the cost of introducing some bias. Formally, the function $h$ needs to satisfy the condition (**C1**), that is *defined* by

$$h \text{ satisfies } (\textbf{C1}) \iff \forall (p, q, c) \in [0,1]^2 \times [-1, 0], \quad pc/q \leq h(p, q, c) \leq 0. \tag{C1}$$

With such function $h$, the regularized IPS estimator reads

$$\hat{R}_n^h(\pi) = \frac{1}{n} \sum_{i=1}^{n} h\left(\pi(a_i|x_i), \pi_0(a_i|x_i), c_i\right) = \frac{1}{n} \sum_{i=1}^{n} h_i \, , \tag{3}$$

where $h_i = h\left(\pi(a_i|x_i), \pi_0(a_i|x_i), c_i\right)$. We recover standard IPS in (2) when $h(p, q, c) = pc/q$. Numerous regularization functions $h$ were studied in the literature. For example,

$$h(p, q, c) = \min(p/q, M)c \, , M \in \mathbb{R}^+ \implies \text{Clipping [10] } , \tag{4}$$
$$h(p, q, c) = pc/q^\alpha \, , \alpha \in [0, 1] \implies \text{Exponential Smoothing [5] } ,$$
$$h(p, q, c) = pc/(q + \gamma) \, , \gamma \geq 0 \implies \text{Implicit Exploration [21] } .$$

Other IW regularizations include Harmonic [38] and Shrinkage [54]. With $h$ satisfying (**C1**), we can derive our core result: a family of high-probability bounds that hold for regularized IPS.

## 3 Pessimistic off-policy evaluation

Standard OPE directly uses estimates of the risk, without capturing their associated uncertainty. This limits its effectiveness in critical applications. Pessimistic OPE addresses this issue by relying on finite sample, high-probability upper bounds to assess any policy's worst-case risk [15, 32]. This section contributes to this effort and focuses on providing novel, finite sample, tight upper bounds on the risk. This is achieved by deriving general bounds applicable to regularized IPS in (3).

### 3.1 Preliminaries and unified risk bounds

Let $\lambda > 0$, $\pi \in \Pi$, and $h$ satisfying (**C1**), we define

$$\hat{\mathcal{M}}_n^{h,\ell}(\pi) = \frac{1}{n} \sum_{i=1}^{n} h_i^\ell \, , \qquad \text{and} \quad \psi_\lambda(x) = \frac{1}{\lambda}\left(1 - \exp(-\lambda x)\right), \, \forall x \in \mathbb{R}, \tag{5}$$

where $\hat{\mathcal{M}}_n^{h,\ell}(\pi)$ is the empirical $\ell$-th moment of regularized IPS $\hat{R}_n^h(\pi)$, and $\psi_\lambda : \mathbb{R} \to \mathbb{R}$ is a *contraction* function satisfying $\psi_\lambda(x) \leq x$ for any $x \in \mathbb{R}$. Then, we state our first result.

**Proposition 1** (Empirical moments risk bound). *Let $\pi \in \Pi$, $L \geq 1$, $\delta \in (0, 1]$, $\lambda > 0$, and $h$ satisfying (**C1**). Then it holds with probability at least $1 - \delta$ that*

$$R(\pi) \leq U_L^{\lambda,h}(\pi) \, , \quad \text{with} \quad U_L^{\lambda,h}(\pi) = \psi_\lambda\left(\hat{R}_n^h(\pi) + \sum_{\ell=2}^{2L} \frac{\lambda^{\ell-1}}{\ell} \hat{\mathcal{M}}_n^{h,\ell}(\pi) + \frac{\ln(1/\delta)}{\lambda n}\right), \tag{6}$$

*where $\psi_\lambda$ and $\hat{\mathcal{M}}_n^{h,\ell}(\pi)$ are both defined in (5), and recall that $\psi_\lambda(x) \leq x$.*

In Appendix F.1, we provide detailed proof, leveraging Chernoff bounds with a careful analysis of the moment-generating function. This results in the first empirical, high-order moment bound for offline contextual bandits, with several advantages. First, the bound applies to any regularization function $h$ that satisfies the mild condition (C1), enabling the design of a tailored $h$ that minimizes the bound. Second, it relies solely on empirical moments, without assuming the existence of theoretical moments. Third, the bound is fully empirical and tractable, facilitating efficient implementation of pessimism. Lastly, the parameter $L$ controls the number of moments used, allowing a balance between bound tightness and computational cost. Specifically, for sufficiently small values of $\lambda$, higher values of $L$ yield tighter bounds, though potentially at the cost of increased computational complexity as we would need to compute higher order moments. This is formally stated as follows.

**Proposition 2** (Impact of $L$). *Let $\pi \in \Pi$, $\delta \in (0, 1]$, $\lambda > 0$, $L \geq 1$, and $h$ satisfying* (C1). *Then,*

$$\lambda \leq \min_{i \in [n]} \left\{ \frac{2L + 2}{(2L + 1)|h_i|} \right\} \implies U_{L+1}^{\lambda,h}(\pi) \leq U_L^{\lambda,h}(\pi). \tag{7}$$

From (7), the bound $U_L^{\lambda,h}(\pi)$ in (6) becomes a decreasing function of $L$ when $\lambda \leq \min_{i \in [n]}(1/|h_i|)$, suggesting that for sufficiently small $\lambda$, the tightest bound is achieved as $L \to \infty$. This condition on $\lambda$ also depends on the values of $h$, highlighting the importance of the regularizer choice $h$. In fact, once we evaluate our bounds at their optimal regularizer function $h$, this condition on $\lambda$ becomes unnecessary when comparing some of the optimal bounds. Specifically, we demonstrate in the following proposition that the bound with $L = 1$ can be always improved by increasing $L$.

**Proposition 3** (Comparison of our bounds). *Let $\pi \in \Pi$, and $\lambda > 0$, we define*

$$U_L^\lambda(\pi) = \min_h U_L^{\lambda,h}(\pi), \quad \text{and} \quad h_{*,L} = \operatorname*{argmin}_h U_L^{\lambda,h}(\pi), \tag{8}$$

*with the minimum taken over $h$ satisfying* (C1). *Then, for any $\lambda > 0$, it holds that for any $L > 1$, $U_L^\lambda(\pi) \leq U_1^\lambda(\pi)$. In particular, for any $\lambda > 0$,*

$$U_\infty^\lambda(\pi) \leq U_1^\lambda(\pi). \tag{9}$$

Proposition 3 shows that, irrespective of the value of $\lambda$, the bound with $L = 1$ can be always improved by bounds of increased moment order $L$, evaluated at their optimal regularizer $h_{*,L}$. This result encourages us to study bounds with high moment order $L$, especially if we can derive their optimal regularizers $h_{*,L}$. To this end, we examine two cases: $L = 1$, which results in an empirical second-moment bound, and $L \to \infty$, yielding a tight bound that does not require computing high-order moments. For each case, we identify the function $h$ that minimizes the bound. If the minimizer for $L = 1$ is a variant of the clipping estimator [10], minimizing $L \to \infty$ motivates a novel logarithmic smoothing estimator. We begin by analyzing our empirical moment risk bound at $L = 1$.

## 3.2 Global clipping

**Corollary 4** (Empirical second-moment risk bound with $L = 1$). *Let $\pi \in \Pi$, $\delta \in (0, 1]$, $\lambda > 0$, and $h$ satisfying* (C1). *Then it holds with probability at least $1 - \delta$ that*

$$R(\pi) \leq \psi_\lambda \left( \hat{R}_n^h(\pi) + \frac{\lambda}{2} \hat{\mathcal{M}}_n^{h,2}(\pi) + \frac{\ln(1/\delta)}{\lambda n} \right). \tag{10}$$

This is a direct consequence of (6) when $L = 1$. The bound holds for any $h$ satisfying (C1). Thus we search for a function $h_{*,1}$ that minimizes bound in (10). This function $h_{*,1}$ writes

$$h_{*,1}(p, q, c) = -\min(p|c|/q, 1/\lambda). \tag{11}$$

In particular, if we assume that costs are binary, $c \in \{-1, 0\}$, then $h_{*,1}$ corresponds to clipping in (4) with parameter $M = 1/\lambda$. This is because $-\min(|c|p/q, 1/\lambda) = \min\left(p/q, \frac{1}{\lambda}\right) c$ when $c$ is binary. This motivates the widely used clipping estimator [10]. However, this also suggests that the standard way of clipping (as in (4)) is only optimal[1] for binary costs. In general, the cost should also be clipped (as in (11)). Finally, with a suitable choice of $\lambda = \mathcal{O}(1/\sqrt{n})$, our bound in Corollary 4, using clipping (i.e., $h = h_{*,1}$), outperforms the existing empirical Bernstein bound [55], which was specifically derived for clipping. This confirms the strength of our general bound, as minimizing it results in a bound with tighter concentration than specialized bounds. Appendix F.4 gives the the proof to find $h_{*,1}$ and formal comparisons with empirical Bernstein are provided in Appendix F.5. In the next section, we study our general bound when we set $L \to \infty$.

---

[1] Here, optimality of a function $h$ is defined with respect to our bound with $L = 1$ (Corollary 4).

### 3.3 Logarithmic smoothing

**Corollary 5** (Empirical infinite-moment bound with $L \to \infty$). *Let $\pi \in \Pi$, $\delta \in (0, 1]$, $\lambda > 0$, and $h$ satisfying (C1). Then it holds with probability at least $1 - \delta$ that*

$$R(\pi) \leq \psi_\lambda\Big( -\frac{1}{n}\sum_{i=1}^n \frac{1}{\lambda} \log\left(1 - \lambda h_i\right) + \frac{\ln(1/\delta)}{\lambda n} \Big). \tag{12}$$

Appendix F.6 provides detailed proof. Setting $L \to \infty$ in (6) results in the bound in Corollary 5, which has different properties than Corollary 4. The resulting bound has a simple expression that does not require computing high order moments. This means that we can obtain the best of both worlds, a tight concentration bound with no additional computational complexity. As the bound is increasing in $h$, the function $h_{*,\infty}$ that minimizes this bound is $h_{*,\infty}(p, q, c) = pc/q$. This corresponds to the standard IPS in (2). This differs from the $L = 1$ bound in Corollary 4 that favored clipping. This shows the impact of the moment order $L$ on the optimal function $h$. For any $\pi \in \Pi$, applying the bound in Corollary 5 with the optimal $h_{*,\infty}$ leads to $U_\infty^\lambda(\pi)$, of the following expression:

$$U_\infty^\lambda(\pi) = \psi_\lambda\Big( \hat{R}_n^\lambda(\pi) + \frac{\ln(1/\delta)}{\lambda n} \Big). \tag{13}$$

Even if we set $h_{*,\infty}(p, q, c) = pc/q$ (without IW regularization), $U_\infty^\lambda(\pi)$ can be seen as a risk upper bound of a novel regularized IPS estimator (satisfying (C1)), called Logarithmic Smoothing (LS):

$$\hat{R}_n^\lambda(\pi) = -\frac{1}{n}\sum_{i=1}^n \frac{1}{\lambda} \log\left(1 - \lambda w_\pi(x_i, a_i)c_i\right). \tag{14}$$

The LS estimator in (14) is defined for any non-negative $\lambda \geq 0$, with its bound in (13) holding for any positive $\lambda > 0$. Notably, $\lambda = 0$ retrieves the standard IPS estimator in (2), while $\lambda > 0$ introduces a bias-variance trade-off by logarithmically smoothing the IWs (Figure 1). This estimator acts as a soft, differentiable variant of clipping with parameter $1/\lambda$. A Taylor expansion of our estimator around $\lambda = 0$ yields

$$\hat{R}_n^\lambda(\pi) = \hat{R}_n(\pi) + \sum_{\ell=2}^\infty \frac{\lambda^{\ell-1}}{\ell}\Big( \frac{1}{n}\sum_{i=1}^n (w_\pi(x_i, a_i)c_i)^\ell \Big).$$

Figure 1: LS with different $\lambda$s.

Thus, LS is a pessimistic estimator *by design*, implicitly implementing a form of *Sample All Moments Penalization*, which generalizes the *Sample Variance Penalization* [55]. To examine the statistical properties of our estimator, we introduce

$$\mathcal{S}_\lambda(\pi) = \mathbb{E}\left[ \frac{(w_\pi(x, a)c)^2}{(1 - \lambda w_\pi(x, a)c)} \right], \tag{15}$$

which quantifies the discrepancy between $\pi$ and $\pi_0$. Notably, $\mathcal{S}_\lambda(\pi)$ is always smaller than the second moment of the IW, effectively interpolating between a weighted first moment ($\lambda \gg 1$) and the second moment ($\lambda = 0$) of IPS. This quantity $\mathcal{S}_\lambda$ characterizes the concentration properties of the LS estimator akin to the coverage ratio for IX estimator [21]. With $\mathcal{S}_\lambda$ defined, we proceed by bounding the mean squared error (MSE) of our estimator, specifically bounding its bias and variance.

**Proposition 6** (Bias-variance trade-off). *Let $\pi \in \Pi$ and $\lambda \geq 0$. Let $\mathcal{B}^\lambda(\pi)$ and $\mathcal{V}^\lambda(\pi)$ be respectively the bias and the variance of the LS estimator. Then we have that*

$$0 \leq \mathcal{B}^\lambda(\pi) \leq \lambda \mathcal{S}_\lambda(\pi), \quad and \quad \mathcal{V}^\lambda(\pi) \leq \frac{\mathcal{S}_\lambda(\pi)}{n}.$$

*Moreover, it holds that for any $\lambda > 0$, the variance is finite as $\mathcal{V}^\lambda(\pi) \leq |R(\pi)|/\lambda n \leq 1/\lambda n$.*

We observe that both the bias and variance are controlled by $\mathcal{S}_\lambda(\pi)$. Particularly, $\lambda = 0$ recovers the IPS estimator in (2), with zero bias and a variance bounded by $\mathbb{E}\left[ w^2(x, a)c^2 \right]/n$. When $\lambda > 0$, a bias-variance trade-off emerges. The bias is always non-negative and is capped at $\lambda \mathcal{S}_\lambda(\pi)$, which diminishes to zero when $\lambda$ is small and goes to $|R(\pi)|$ as $\lambda$ increases. Conversely, the variance decreases with a higher $\lambda$. Notably, $\lambda > 0$ ensures finite variance bounded by $1/\lambda n$, despite the estimator being unbounded. This is different from previous estimators that relied on bounded functions to ensure finite variance. We also prove in the following that a good choice of $\lambda = \mathcal{O}(1/\sqrt{n})$ ensures that our LS estimator enjoys a sub-Gaussian concentration [38].

**Proposition 7** (Sub-Gaussianity and comparison with Metelli et al. [38]). *Let* $\pi \in \Pi$, $\delta \in (0, 1]$ *and* $\lambda > 0$. *Then the following inequalities holds with probability at least* $1 - \delta$:

$$R(\pi) - \hat{R}_n^\lambda(\pi) \leq \frac{\ln(2/\delta)}{\lambda n}, \qquad and \qquad \hat{R}_n^\lambda(\pi) - R(\pi) \leq \lambda \mathcal{S}_\lambda(\pi) + \frac{\ln(2/\delta)}{\lambda n}.$$

*In particular, setting* $\lambda = \lambda_* = \sqrt{\ln(2/\delta)/n\mathbb{E}\left[w_\pi(x,a)^2 c^2\right]}$ *yields that*

$$|R(\pi) - \hat{R}_n^{\lambda_*}(\pi)| \leq \sqrt{2\sigma^2 \ln(2/\delta)}, \qquad where \ \sigma^2 = 2\mathbb{E}\left[w_\pi(x,a)^2 c^2\right]/n. \qquad (16)$$

Thus, a particular choice of $\lambda^*$ ensures that $\hat{R}_n^{\lambda_*}(\pi)$ is sub-Gaussian, with a variance proxy $\sigma^2$ that improves on that obtained for the Harmonic estimator of Metelli et al. [38]. We refer the interested reader to Appendix E.2 for further discussions and proofs.

Next, we focus on the tightness of the LS upper bound in (13) as it will motivate our selection and learning strategies. Proposition 3 already showed that $U_\infty^\lambda(\pi)$, the bound of LS is tighter than $U_1^\lambda(\pi)$, the bound in Corollary 4 evaluated at the Global clipping function $h_{*,1}$. In this section, we compare the LS bound to the already tight IX bound presented by Gabbianelli et al. [21] and demonstrate in the following that the LS bound dominates it in all scenarios.

**Proposition 8** (Comparison with IX of Gabbianelli et al. [21]). *Let* $\pi \in \Pi$, $\delta \in ]0, 1]$ *and* $\lambda > 0$, *the* IX *bound from [21] states that we have with probability at least* $1 - \delta$

$$R(\pi) \leq \hat{R}_n^{\lambda\text{-IX}}(\pi) + \frac{\ln(1/\delta)}{\lambda n}, \quad with \quad \hat{R}_n^{\lambda\text{-IX}}(\pi) = \frac{1}{n}\sum_{i=1}^n \frac{\pi(a_i|x_i)}{\pi_0(a_i|x_i) + \lambda/2}c_i. \qquad (17)$$

*Let* $U_{\text{IX}}^\lambda(\pi)$ *be the upper bound of* (17), *we have for any* $\lambda > 0$:

$$U_\infty^\lambda(\pi) \leq U_{\text{IX}}^\lambda(\pi). \qquad (18)$$

This result states that no matter the scenario, for any evaluated policy $\pi$, and any chosen $\lambda > 0$, the LS bound will be always tighter than IX. The gap between the LS and IX bounds increases when $n$ is small, or when the evaluated policy $\pi$ is stochastic, as demonstrated and developed in Appendix F.8. These findings further validate the effectiveness of our approach, enabling us to identify the LS estimator, with an empirical bound that improves upon the tightest existing bounds. Consequently, we leverage the LS bound in the next section to derive our pessimistic OPS and OPL strategies.

## 4 Off-policy selection and learning

### 4.1 Off-policy selection

Let $\Pi_s = \{\pi_1, ..., \pi_m\}$ be a finite set of policies. In OPS, the goal is to find $\pi_*^s \in \Pi_s$ that satisfies

$$\pi_*^s = \underset{\pi \in \Pi_s}{\operatorname{argmin}} \, R(\pi) = \underset{k \in [m]}{\operatorname{argmin}} \, R(\pi_k). \qquad (19)$$

As we do not have access to the true risk, we use a data-driven selection strategy that guarantees the identification of policies of performance close to that of $\pi_*^s$. Precisely, for $\lambda > 0$, we search for

$$\hat{\pi}_n^s = \underset{\pi \in \Pi_s}{\operatorname{argmin}} \, \hat{R}_n^\lambda(\pi) = \underset{k \in [m]}{\operatorname{argmin}} \, \hat{R}_n^\lambda(\pi_k). \qquad (20)$$

To derive our strategy in (20), we minimize the bound of LS in (13), employing pessimism [27]. Fortunately, in our case, this boils down to minimizing $\hat{R}_n^\lambda(\pi)$, since the other terms in the bound are independent of the target policy $\pi$. This allows us to avoid computing complex statistics [55, 32] and does not require access to the behavior policy $\pi_0$. As we show next, it also ensures low suboptimality.

**Proposition 9** (Suboptimality of our selection strategy in (20)). *Let* $\lambda > 0$ *and* $\delta \in (0, 1]$. *Then, it holds with probability at least* $1 - \delta$ *that*

$$0 \leq R(\hat{\pi}_n^s) - R(\pi_*^s) \leq \lambda \mathcal{S}_\lambda(\pi_*^s) + \frac{2\ln(2|\Pi_s|/\delta)}{\lambda n}, \qquad (21)$$

*where* $\mathcal{S}_\lambda(\pi)$, $\pi_*^s$ *and* $\hat{\pi}_n^s$ *are defined in* (15), (19) *and* (20).

The derived suboptimality bound only requires coverage of the optimal actions (support of the optimal policy $\pi_*^s$), and improves on IX suboptimality [21], matching the minimax suboptimality lower bound of pessimistic methods [34, 27, 28]. Appendix G.1 provides proof of this suboptimality bound, and we discuss how this suboptimality improves upon existing strategies in Appendix E.3. By selecting $\lambda_n^s = \sqrt{2\ln(2|\Pi_S|/\delta)/n}$ for LS, we achieve a suboptimality scaling of $\mathcal{O}(1/\sqrt{n})$,

$$0 \leq R(\hat{\pi}_n^S) - R(\pi_*^S) \leq \left(1 + \mathcal{S}_{\lambda_n^s}(\pi_*^S)\right)\sqrt{2\ln(2|\Pi_S|/\delta)/n}, \tag{22}$$

which ensures finding the optimal policy with sufficient samples. Additionally, the multiplicative constant is smaller when $\pi_0$ is close to $\pi_*^S$, confirming the known observation that it is easier to identify the best policy if it is similar to the behavior policy $\pi_0$.

## 4.2 Off-policy learning

Similar to how we extended the evaluation bound in Corollary 5 (which applies to a single fixed target policy) to OPS (where it applies to a finite set of target policies), we can further derive bounds for an infinite policy class $\Pi$, enabling OPL. Several approaches have been proposed in previous work, primarily based on replacing the finite union bound over policies with more sophisticated uniform-convergence arguments. This was used by [55], which derived a variance-sensitive bound scaling with the covering number [61]. Since these approaches incorporate a complexity term that depends only on the policy class, the resulting pessimistic learning strategy (which minimizes the upper bound) would be similar to the selection strategy adopted earlier, leading, for a fixed $\lambda$, to

$$\hat{\pi}_n^L = \underset{\pi \in \Pi}{\arg\min}\, \hat{R}_n^\lambda(\pi) + \frac{\mathcal{C}(\Pi)}{\lambda n} = \underset{\pi \in \Pi}{\arg\min}\, \hat{R}_n^\lambda(\pi). \tag{23}$$

where $\mathcal{C}(\Pi)$ is a complexity measure [61]. This learning strategy is straightforward because it involves a smooth estimator that can be optimized using first-order methods and does not require second-order statistics. However, analyzing this approach is more challenging because the complexity measure $\mathcal{C}(\Pi)$ varies depending on the policy class considered, is often intractable [49] and can only be upper bounded with problem dependent constants [28].

Instead of the method described above, we derive PAC-Bayesian generalization bounds [37, 11] that apply to arbitrary policy classes. This framework has been shown to provide strong performance guarantees for OPL in practical scenarios [49, 5]. The PAC-Bayesian framework analyzes the performance of policies by viewing them as randomized predictors [35]. Specifically, let $\mathcal{F}(\Theta) = \{f_\theta : \mathcal{X} \to [K], \theta \in \Theta\}$ be a set of parameterized predictors that associate the context $x$ with the action $f_\theta(x) \in [K]$. Let $\mathcal{P}(\Theta)$ be the set of all probability distributions on $\Theta$. Each distribution $Q \in \mathcal{P}(\Theta)$ defines a policy $\pi_Q$ by setting the probability of action $a$ given context $x$ as the probability that a random predictor $f_\theta \sim Q$ maps $x$ to action $a$, that is,

$$\pi_Q(a|x) = \mathbb{E}_{\theta \sim Q}\left[\mathbb{1}\left[f_\theta(x) = a\right]\right], \qquad \forall(x,a) \in \mathcal{X} \times \mathcal{A}. \tag{24}$$

This characterization is not restrictive as any policy can be represented in this form [49]. Deriving PAC-Bayesian generalization bounds with this policy definition requires the regularized IPS to be linear in the target policy $\pi$ [35, 5, 21]. Our estimator LS in (14) is non-linear in $\pi$. Therefore, for this PAC-Bayesian analysis, we introduce a linearized variant of LS, called LS-LIN, and defined as

$$\hat{R}_n^{\lambda\text{-LIN}}(\pi) = -\frac{1}{n}\sum_{i=1}^n \frac{\pi(a_i|x_i)}{\lambda}\log\left(1 - \frac{\lambda c_i}{\pi_0(a_i|x_i)}\right), \tag{25}$$

which smooths the impact of the behavior propensity $\pi_0$ instead of the IWs $\pi/\pi_0$. We provide in the following a core result of this section, the PAC-Bayesian bound that defines our learning strategy.

**Proposition 10** (PAC-Bayes learning bound for $\hat{R}_n^{\lambda\text{-LIN}}$). *Given a prior $P \in \mathcal{P}(\Theta)$, $\delta \in (0,1]$ and $\lambda > 0$, the following holds with probability at least $1 - \delta$:*

$$\forall Q \in \mathcal{P}(\Theta), \quad R(\pi_Q) \leq \psi_\lambda\left(\hat{R}_n^{\lambda\text{-LIN}}(\pi_Q) + \frac{\mathcal{KL}(Q||P) + \ln\frac{1}{\delta}}{\lambda n}\right), \tag{26}$$

*where $\mathcal{KL}(Q||P)$ is the Kullback-Leibler divergence from $P$ to $Q$.*

PAC-Bayes bounds hold uniformly for all distributions $Q \in \mathcal{P}(\Theta)$ and replace the complexity measure $\mathcal{C}(\Pi)$ with the divergence $\mathcal{KL}(Q||P)$ from a reference *prior* distribution $P$. Extensive research focuses on identifying the best strategies for choosing this prior $P$ [40]. While these bounds hold for any fixed prior $P$, in practice, it is typically set to the distribution inducing the behavior policy $\pi_0$, meaning $P$ satisfies $\pi_0 = \pi_P$. This leads to an intuitive learning principle: by minimizing the upper bound, we seek policies with good empirical risk that do not deviate significantly from $\pi_0$.

Our bound can also be obtained using the truncation method from Alquier [1, Corollary 2.5]. This bound surpasses the already tight PAC-Bayesian bounds derived for Clipping [49], Exponential Smoothing [5], and Implicit Exploration [21], resulting in the tightest known generalization bound in OPL. Appendix G.2 gives formal proof of this bound and comparisons with existing PAC-Bayesian bounds can be found in Appendix E.4. For a fixed $\lambda$ and a fixed prior $P$, we derive a learning strategy that minimizes the upper bound for a subset $\mathcal{L}(\Theta) \subseteq \mathcal{P}(\Theta)$ of distributions, seeking

$$Q_n = \underset{Q \in \mathcal{L}(\Theta)}{\operatorname{argmin}} \left\{ \hat{R}_n^{\lambda\text{-LIN}}(\pi_Q) + \frac{\mathcal{KL}(Q||P)}{\lambda n} \right\}, \quad \text{and setting } \hat{\pi}_n^{\text{L}} = \pi_{Q_n}. \tag{27}$$

(27) is tractable and can be efficiently optimized for various policy classes [49, 5]. Below, we analyze its suboptimality compared to the best policy in the chosen class, $\pi_{Q^*} = \operatorname{argmin}_{Q \in \mathcal{L}(\Theta)} R(\pi_Q)$.

**Proposition 11** (Suboptimality of the learning strategy in (27)). *Let $\lambda > 0$, $P \in \mathcal{L}(\Theta)$ and $\delta \in (0, 1]$. Then, it holds with probability at least $1 - \delta$ that*

$$0 \le R(\hat{\pi}_n^{\text{L}}) - R(\pi_{Q^*}) \le \lambda \mathcal{S}_\lambda^{\text{LIN}}(\pi_{Q^*}) + \frac{2\left(\mathcal{KL}(Q^*||P) + \ln(2/\delta)\right)}{\lambda n}, \tag{28}$$

*where $\mathcal{S}_\lambda^{\text{LIN}}(\pi) = \mathbb{E}\left[\pi(a|x)c^2/(\pi_0^2(a|x) - \lambda\pi_0(a|x)c)\right]$ and $\hat{\pi}_n^{\text{L}}$ is defined in (27).*

Our suboptimality bound only requires coverage of the support of the optimal policy $\pi_{Q_*}$. This bound matches the minimax suboptimality lower bound of pessimistic learning with deterministic policies [28]. Appendix G.3 provides a proof of Proposition 11, while Appendix E.5 discusses the suboptimality bound further and proves that it improves on the IX learning strategy of [21, Section 5]. Setting $\lambda_n^l = 2/\sqrt{n}$ guarantees us a suboptimality that scales with $\mathcal{O}(1/\sqrt{n})$ as

$$0 \le R(\hat{\pi}_n^{\text{L}}) - R(\pi_{Q^*}) \le (2\mathcal{S}_{\lambda_n^l}^{\text{LIN}}(\pi_{Q^*}) + \mathcal{KL}(Q^*||P) + \ln(2/\delta))/\sqrt{n}.$$

By setting the reference $P$ to the distribution inducing $\pi_0$, we find that the learning suboptimality is reduced when the behavior policy $\pi_0$ is close to the optimal policy $\pi_{Q^*}$. This is similar to the suboptimality for our selection strategy. The suboptimality upper bound reflects a common intuition in the OPL literature: pessimistic learning algorithms converge faster when $\pi_0$ is close to $\pi_{Q^*}$.

## 5 Experiments

Our experimental setup follows the standard multiclass-to-bandit conversion used in prior studies [18, 55]. Each multi-class dataset has features and labels and we convert it to contextual bandit problems where contexts correspond to features and actions to labels. Precisely, the reward $r$ for taking action (label) $a$ with context (features) $x$ is modeled as Bernoulli with probability $p_x = \epsilon + \mathbb{1}\left[a = \rho(x)\right](1 - 2\epsilon)$, where $\rho(x)$ be the true label of features $x$, and $\epsilon$ is a noise parameter. In particular, the true label $\rho(x)$ represents the action with the highest average reward for context $x$. This setup ensures an average reward of $1 - \epsilon$ for the optimal action $\rho(x)$ and $\epsilon$ for all others, constructing a logged bandit feedback dataset in the form $\{x_i, a_i, c_i\}_{i \in [n]}$, where $c_i = -r_i$ is the associated cost.

### 5.1 Off-policy evaluation and selection experiments

For both evaluation and selection, we adopt the same experimental design as [32] to facilitate the comparison. We consider exponential target policies $\pi(a|x) \propto \exp(\frac{1}{\tau}f(a, x))$, with $\tau$ a temperature controlling the policy's entropy and $f(a, x)$ the score of the item $a$ for the context $x$. We use this to define ideal policies as $\pi^{\text{ideal}}(a|x) \propto \exp(\frac{1}{\tau}\mathbb{I}\{\rho(x) = a\})$, and also create faulty, mismatching policies for which the peak is shifted to another, wrong action for a set of faulty actions $F \subset [K]$. To recreate real world scenarios, we also consider policies directly learned from logged bandit feedback, of the form $\pi_{\theta^{\text{IPS}}}(a|x) \propto \exp(\frac{1}{\tau}x^t\theta_a^{\text{IPS}})$ and $\pi_{\theta^{\text{SN}}}(a|x) \propto \exp(\frac{1}{\tau}x^t\theta_a^{\text{SN}})$, with their parameters learned

Table 1: Bound's tightness ($|U(\pi)/R(\pi) - 1|$) with varying number of samples of the `kropt` dataset.

| Number of samples | SN-ES | cIPS-EB | IX | cIPS-L=1 (Ours) | LS (Ours) |
|---|---|---|---|---|---|
| $2^8$ | 1.000 | 0.917 | 0.373 | 0.364 | **0.362** |
| $2^9$ | 1.000 | 0.732 | 0.257 | 0.289 | **0.236** |
| $2^{10}$ | 0.794 | 0.554 | 0.226 | 0.240 | **0.213** |
| $2^{11}$ | 0.649 | 0.441 | 0.171 | 0.197 | **0.159** |
| $2^{12}$ | 0.472 | 0.327 | 0.126 | 0.147 | **0.117** |
| $2^{13}$ | 0.374 | 0.204 | 0.062 | 0.077 | **0.054** |
| $2^{14}$ | 0.257 | 0.138 | 0.041 | 0.049 | **0.035** |

by respectively minimizing the IPS [24] and SN [56] empirical risks. More details on the definition of the different policies are given in Appendix H. Finally, 11 real multiclass classification datasets are chosen from the UCI ML Repository [8] (See Table 3 in Appendix H.1.1) with various number of samples, dimensions and action space sizes to conduct our experiments[2].

**(OPE) Tightness of the bounds.** Evaluating the worst case performance of a policy is done through evaluating risk upper bounds [10, 32]. This means that a better evaluation will solely depend on the tightness of the bounds used. To this end, given a policy $\pi$, we are interested in bounds $U(\pi)$ with a small relative radius $|U(\pi)/R(\pi) - 1|$. We compare our newly derived bounds (cIPS-L=1 for $U_1^\lambda$ and LS for $U_\infty^\lambda$ both with $\lambda = 1/\sqrt{n}$) to empirical evaluation bounds of the literature: SN-ES: the Efron Stein bound for Self Normalized IPS [32], cIPS-EB: Empirical Bernstein for Clipping [55] and the recent IX: Implicit Exploration bound [21]. The first experiment uses the `kropt` dataset with $\epsilon = 0.2$, collects bandit feedback with faulty behavior policy (with $\tau = 0.25$) to evaluate an ideal policy ($\tau = 0.1$), and explores how the relative radiuses of the considered bounds shrink while varying the number of datapoints. Table 1 compiles the results of the experiments and suggest that the LS bound is tighter than its competitors no matter the size of the feedback collected. The second experiments uses all 11 datasets, with different behavior policies ($\tau_0 \in \{0.2, 0.25, 0.3\}$) and different noise levels ($\epsilon \in \{0., 0.1, 0.2\}$) to evaluate ideal policies with different temperatures ($\tau \in \{0.1, 0.2, 0.3, 0.4, 0.5\}$), defining $\sim 500$ different scenarios to validate our findings. We plot in Figure 2 the cumulative distribution of the relative radius of the considered bounds. We observe that while cIPS-L=1 and IX can be comparable, the LS bound is tighter than all its competitors. We also provide detailed results in Appendix H.1.2 that further confirm the superiority of the LS bound.

**(OPS) Find the best, avoid the worst policy.** Policy selection aims at identifying the best policy among a set of finite candidates. In practice, we are interested in finding policies that improve on $\pi_0$ and avoid policies that perform worse than $\pi_0$. To replicate real world scenarios, we design an experiment where $\pi_0$ is a faulty policy ($\tau = 0.2$), that collects noisy ($\epsilon = 0.2$) interaction data, some of which is used to learn $\pi_{\theta^{\text{IPS}}}, \pi_{\theta^{\text{SN}}}$, and that we add to our discrete set of policies $\Pi_{k=4} = \{\pi_0, \pi^{\text{ideal}}, \pi_{\theta^{\text{IPS}}}, \pi_{\theta^{\text{SN}}}\}$. The goal is to measure the ability of our selection strategies to choose from $\Pi_{k=4}$, better performing policies than $\pi_0$. We thus define three possible outcomes: a strategy can select *worse* performing policies, *better* performing or the *best* policy. Our goal in these experiments is to empirically validate the pitfalls of point estimators while confirming the benefits of using the pessimism principle. To this end, we compare *pessimistic* selection strategies to policy selection using the classical point estimators IPS [24] and SN [56]. The comparison is conducted on the 11 UCI datasets with 10 different seeds resulting in 110 scenarios. We plot in Figure 2 the percentage of time each method selected the best policy, a better or a worse policy than $\pi_0$. While risk estimators can identify the best policy, they are unreliable as they can choose worse performing policies than $\pi_0$, a catastrophic outcome in critical applications. Pessimistic selection is more conservative, as it avoids poor performing policies completely and empirically confirms that tighter upper bounds result in better selection strategies: LS upper bound is less conservative and finds best policies the most (comparable to SN) while never selecting poor performing policies. Fine grained results (for each dataset) can be found in Appendix H.1.3.

## 5.2 Off-policy learning experiments

We follow the successful off policy learning paradigm based on directly minimizing PAC-Bayesian risk generalization bounds [49, 5] as it comes with guarantees of improvement and avoids hyper-

---

[2]The code can be found at https://github.com/otmhi/offpolicy_ls.

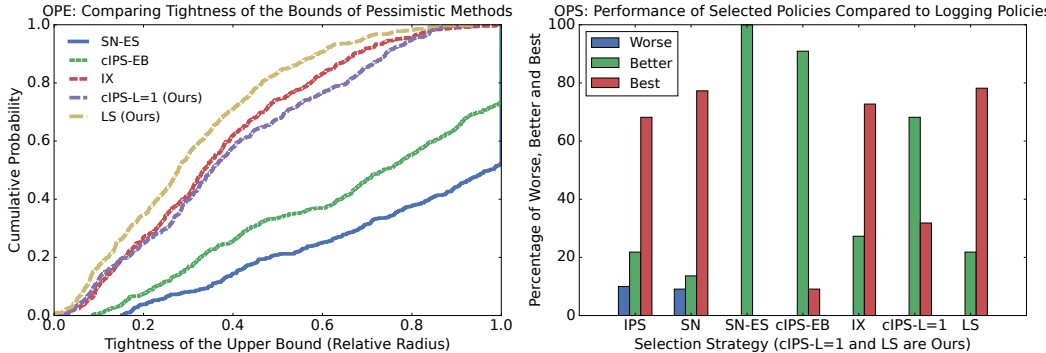

Figure 2: Results for OPE and OPS experiments.

|  | cIPS | cvcIPS | ES | IX | LS-LIN (Ours) |
|---|---|---|---|---|---|
| $rI(U(\hat{\pi}_n^{\text{L}}))$ | 14.48% | 21.28% | 7.78% | _24.74%_ | **26.31%** |
| $rI(R(\hat{\pi}_n^{\text{L}}))$ | 28.13% | _33.64%_ | 29.44% | **36.70%** | **36.76%** |

Table 2: OPL: Relative Improvement of guaranteed risk and true risk averaged over 200 scenarios.

parameter tuning. For comparable results, we use the same 4 datasets (described in Appendix H.2, Table 7) as in [49, 5] and adopt the **LGP**: Linear Gaussian Policies [49] as our class of parametrized policies. For each dataset, we use behavior policies trained on a small fraction of the data in a supervised fashion, combined with different inverse temperature parameters $\alpha \in \{0.1, 0.3, 0.5, 0.7, 1.\}$ to cover cases of diffused and peaked behavior policies. These policies generate for 10 different seeds, 10 logged bandit feedback datasets resulting in 200 different scenarios to test our learning approaches. In the PAC-Bayesian OPL paradigm, we minimize the empirical upper bounds $U(\pi)$ directly and obtain the learned policy as the bound's minimizer $\hat{\pi}_n^{\text{L}}$ (as in (27)). With $\hat{\pi}_n^{\text{L}}$ obtained, we are interested in two quantities: The guaranteed risk by the bound, which is the value of the bound $U(\hat{\pi}_n^{\text{L}})$ at its minimizer. This quantity reflects the worst case performance of the learned policy, a lower value implies stronger performance guarantees. We are also interested in the true risk of the minimizer of the bound $R(\hat{\pi}_n^{\text{L}})$ as it translates the performance of the obtained policy acting on unseen data. As this learning paradigm is based on optimizing tractable, generalization bounds, we only compare our approach to methods that provide them. Precisely, we compare our LS-LIN learning strategy in (27) to strategies based on minimizing off-policy PAC Bayesian bounds from the literature: clipped IPS (cIPS) and Control Variate clipped IPS (cvcIPS) [49], Exponential Smoothing (ES) [5] and Implicit Exploration (IX) [21]. The results are summarized in Table 2 where we compute:

$$rI(x) = (R(\pi_0) - x)/(R(\pi_0) - R(\pi^*)) = (R(\pi_0) - x)/(R(\pi_0) + 1),$$

the improvement over $R(\pi_0)$ achieved by minimizing the different bounds in terms of $x \in \{U, R\}$ (guaranteed risk and true risk respectively), relative to an ideal improvement. This metric helps us normalize the results, and we report its average over 200 different scenarios, with results in bold being significantly better. Fine grained results can be found in Appendix H.2.4. We observe that the LS-LIN PAC-Bayesian bound improves substantially on its competitors in terms of the guaranteed risk, and also obtains the best performing policies (on par with the IX PAC-Bayesian bound).

## 6 Conclusion

Motivated by the *pessimism* principle, we have derived novel, empirical risk upper bounds tailored for the regularized IPS family of estimators. Minimizing these bounds within this family unveiled Logarithmic Smoothing, a simple estimator with good concentration properties. With its tight upper bound, LS confidently evaluates a policy, and shows provably better guarantees for both selecting and learning policies than all competitors. Our upper bounds remain broadly applicable, only requiring *negative costs*. While this condition does not impact importance weighting estimators, it does not hold for doubly robust estimators. Extending our approach to derive empirical bounds for this type of estimators presents a nontrivial, yet interesting task to explore in future work. Another potential extension would be to relax the i.i.d. assumption of the contextual bandit problem to address, the general offline Reinforcement Learning setting. This direction will introduce a more challenging estimation task and requires developing new concentration bounds.

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

# Table of Contents for Supplementary Material

# A   Limitations

This work develops theoretically grounded and practical pessimistic approaches for the offline contextual bandit setting. Even if the proposed algorithms are general, and provably better than competitors, they still suffer from the intrinsic limitations of importance weighting estimators. Specifically, our method, as presented, will perform poorly in *extremely* large action spaces. However, these limitations can be mitigated by incorporating additional structure as in Saito and Joachims [45], Saito et al. [46]. Another limitation arises from the offline contextual bandit setting itself, which assumes i.i.d. observations. While this assumption is valid in simple scenarios, it becomes unsuitable once we want to capture the long term effect of interventions. Extending our results to the more general, reinforcement learning setting would be an interesting research direction as it comes with a challenging estimation task and will require developing new concentration bounds.

# B   Broader impact

Our work contributes to the development of theoretically grounded and practical pessimistic approaches for the offline contextual bandit setting. The derived algorithms can improve the robustness of decision-making processes by prioritizing safety and minimizing uncertainty associated risks. By leveraging pessimistic strategies, we ensure that decisions are made with a conservative bias, thereby potentially improving outcomes in high-stakes environments where the cost of errors is substantial. Although our framework and algorithms have broad, potentially good applications, their specific social impacts will solely depend on the chosen application domain.

# C   Extended related work

**Offline contextual bandits.** Contextual bandit is a widely adopted framework for online learning in uncertain environments [33]. However, some real-world applications present challenges for existing online algorithms, and thus offline methods that leverage historical data to optimize decision-making have gained traction [10]. Fortunately, large datasets summarizing past interactions are often available, allowing agents to improve their policies offline [55]. Our work explores this offline approach, known as offline (or off-policy) contextual bandits [16]. In this setting, off-policy evaluation (OPE) estimates policy performance using historical data, mimicking real-time evaluations. Depending on the application, the goal might be to find the best policy within a predefined finite set (off-policy selection (OPS)) or the optimal policy overall (off-policy learning (OPL)).

**Off-policy evaluation.** In recent years, OPE has experienced a noticeable surge of interest, with numerous significant contributions [16–18, 60, 19, 54, 38, 32, 45, 47, 26]. The literature on OPE can be broadly classified into three primary approaches. The first, referred to as the direct method (DM) [26, 6], involves the development of a model designed to approximate expected costs for any context-action pair. This model is subsequently employed to estimate the performance of the policies. This approach is often designed for specific applications such as large-scale recommender systems [47, 26, 4]. The second approach, known as inverse propensity scoring (IPS) [24, 17], aims to estimate the costs associated with the evaluated policies by correcting for the inherent preference bias of the behavior policy within the dataset. While IPS maintains its unbiased nature when operating under the assumption that the evaluation policy is absolutely continuous with respect to the behavior policy, it can be susceptible to high variance and substantial bias when this assumption is violated [43]. In response to the variance issue, various techniques have been introduced, including clipping [25, 10], shrinkage [54], power-mean correction [38], implicit exploration [21], self-normalization [56], among others [22]. The third approach, known as doubly robust (DR) [42, 9, 16, 18, 19], combines elements from both the direct method (DM) and inverse propensity scoring (IPS). This work focuses on regularized IPS.

**Off-policy selection and learning.** as in OPE, three key approaches dominate: DM, IPS and DR in OPS and OPL. In OPS, all these methods share the same core objective: identifying the policy with the highest estimated reward from a finite set of candidates. However, they differ in their reward estimation techniques, as discussed in the OPE section above. In contrast, in OPL, DM either deterministically selects the action with the highest estimated reward or constructs a distribution based on these estimates. IPS and DR, on the other hand, employ gradient descent for policy learning [55], updating a parameterized policy denoted by $\pi_\theta$ as $\theta_{t+1} \leftarrow \theta_t - \nabla_\theta R(\pi_\theta)$ for each iteration $t$.

Since the true risk $R$ is unknown, $\nabla_\theta R(\pi_\theta)$ is unknown and needs to be estimated using techniques like IPS or DR.

**Pessimism in offline contextual bandits.** Most OPE studies directly use their point estimators of the risk in OPE, OPS and OPL. However, point estimators can deviate from the true value of the risk, rendering them unreliable for decision-making. Therefore, and to increase safety, alternative approaches focus on constructing bounds on the risk. These bounds, either asymptotic [10, 48, 15] or finite sample [32, 21], aim to evaluate a policy's worst-case performance, adhering to the principle of *pessimism in face of uncertainty* [27]. The principle of pessimism transcends OPE, influencing both OPS and OPL. In these domains, strategies are predominantly inspired by, or directly derived from, upper bounds on the true risk [55, 35, 32, 49, 5, 59]. Consider OPS: [32] leveraged an Efron-Stein bound for the self-normalized IPS estimator, while [21] anchored their analysis on a bound constructed with the Implicit Exploration estimator. Shifting focus to OPL, [55] combined the empirical Bernstein bound [36] with the clipping estimator, motivating sample variance penalization for policy learning. Recent advancements include modifications to the penalization term [59] to be scalable and efficient.

**PAC-Bayes extension.** The PAC-Bayesian paradigm [37, 11] (see Alquier [2] for a recent introduction) provides a rich set of tools to prove generalization bounds for different statistical learning problems. The classical (online) contextual bandit problem received a lot of attention from the PAC-Bayesian community with the seminal work of Seldin et al. [52]. It is just recently that these tools were adapted to the offline contextual bandit setting, with [35] that introduced a clean and scalable PAC-Bayesian perspective to OPL. This perspective was further explored by [20, 49, 5, 7, 21], leading to the development of tight, tractable PAC-Bayesian bounds suitable for direct optimization.

**Large action space extension.** While regularization techniques can improve IPS properties, they often fall short when dealing with extremely large action spaces. Additional assumptions regarding the structure of the contextual bandit problem become necessary. For example, Saito and Joachims [45] introduced the Marginalized IPS (MIPS) framework and estimator. MIPS leverages auxiliary information about the actions in the form of action embeddings. Roughly speaking, MIPS assumes access to embeddings $e_i$ within logged data and defines the risk estimator as

$$\hat{R}_n^{\text{MIPS}}(\pi) = \frac{1}{n} \sum_{i=1}^n \frac{\pi\left(e_i \mid x_i\right)}{\pi_0\left(e_i \mid x_i\right)} c_i = \frac{1}{n} \sum_{i=1}^n w\left(x_i, e_i\right) c_i\,,$$

where the logged data $\mathcal{D}_n = \{(x_i, a_i, e_i, r_i)\}_{i=1}^n$ now includes action embeddings for each data point. The marginal importance weight

$$w(x, e) = \frac{\pi(e \mid x)}{\pi_0(e \mid x)} = \frac{\sum_a p(e \mid x, a)\pi(a \mid x)}{\sum_a p(e \mid x, a)\pi_0(a \mid x)}$$

is a key component of this approach. Compared to IPS and DR, MIPS achieves significantly lower variance in large action spaces [45] while maintaining unbiasedness if the action embeddings directly influence costs $c$. This necessitates informative embeddings that capture the causal effects of actions on costs. However, high-dimensional embeddings can still lead to high variance for MIPS, similar to IPS. Additionally, high bias can arise if the direct effect assumption is violated and embeddings fail to capture these causal effects. This bias is particularly present when performing action feature selection for dimensionality reduction. Recent work proposes learning such embeddings directly from logged data [41, 44, 14], or loosen this assumption [58, 46]. Our proposed importance weight regularization can be potentially combined with these estimators under their respective assumptions on the underlying structure of the contextual bandit problem, extending our approach to large action spaces, and we posit that this will be beneficial when, for example, the action embedding dimension is high. Another line of research in large action spaces is more interested with the learning problem, precisely solving the optimization issues arising from policies defined on large action spaces. Indeed, naive optimization tends to be slow and scales linearly with the number of actions $K$ [12]. Recent work [51, 50] solve this by leveraging fast maximum inner product search [53, 3] in the training loop, reducing the optimization complexity to *logarithmic* in the action space size. These methods however require a linear objective on the target policy. Luckily, our PAC-Bayesian learning objective is linear in the policy and its optimization is amenable to such acceleration.

**Continuous action space extension.** While research has predominantly focused on discrete action spaces, a limited number of studies have tackled the continuous case [29, 13, 59]. For example, [29]

explored non-parametric evaluation and learning of continuous action policies using kernel smoothing, while [13] investigated the semi-parametric setting. Recently, [59] leveraged the smoothing approach from [31] to extend their discrete OPL method to continuous actions. Our work can either use the densities directly, or be similarly extended to continuous actions through a well-defined discretization of the space. Imagine a scenario with infinitely many actions, where policies are defined by density functions. For any context $x$, $\pi(a \mid x)$ represents the density function that maps actions $a$ to probabilities. The discretization process transforms the original contextual bandit problem characterized by the density-based policy class $\Pi$ into an OPL problem defined by a discrete, mass-based policy class $\Pi_K$ (for a finite number of actions $K$). Each policy within $\Pi_K$ approximates a policy in $\Pi$ through a smoothing process.

## D   Useful lemmas

In the following, and for any quantity $Z$, all expectations are computed w.r.t to the distribution of the data when playing actions under the behaviour policy $\pi_0$, as in:

$$\mathbb{E}\left[Z\right] = \mathbb{E}_{x \sim \nu, a \sim \pi_0(\cdot \mid x), c \sim p(\cdot \mid x, a)}\left[Z\right].$$

A lot of the results derived in the paper are based on the use of the well known Chernoff Inequality, that we state below for a sum of i.i.d. random variables:

---

**Lemma 12** (Chernoff Inequality for a sum of i.i.d. random variables.)**.** *Let $a \in \mathbb{R}$, $n \in \mathbb{N}^*$ and $\{X_i, i \in [n]\}$ a collection of $n$ i.i.d. random variables. The following concentration bounds on the right tail of $\sum_{i \in [n]} X_i$ hold for any $\lambda \geq 0$:*

$$P\left(\sum_{i \in [n]} X_i > a\right) \leq \left(\mathbb{E}\left[\exp\left(\lambda X_1\right)\right]\right)^n \exp(-\lambda a)$$

---

This result is classical in the literature [23] and we omit its proof. We will also need the following lemma, that states the monotonous nature of a key function in our analysis, and that we take the time to prove.

---

**Lemma 13.** *Let $L \geq 1$ and $f_L$ be the following function:*

$$f_L(x) = \frac{\log(1+x) - \sum_{\ell=1}^{L} \frac{(-1)^{\ell-1}}{\ell} x^\ell}{(-1)^L x^{L+1}}.$$

*We have that $f_L$ is a decreasing function in $\mathbb{R}^+$ for all $L \in \mathbb{N}^*$.*

---

*Proof.* Let $L \geq 1$ and $f_L$ be the following function:

$$f_L(x) = \frac{\log(1+x) - \sum_{\ell=1}^{L} \frac{(-1)^{\ell-1}}{\ell} x^\ell}{(-1)^L x^{L+1}}.$$

Let $x \in \mathbb{R}^+$, we have the following identity holding $\forall t > 0$ and $\forall n \geq 0$:

$$\frac{1 + (-1)^n t^{n+1}}{1+t} = \sum_{k=0}^{n} (-1)^k t^k \iff \frac{1}{1+t} = \sum_{k=0}^{n} (-1)^k t^k + \frac{(-1)^{n+1} t^{n+1}}{1+t}. \tag{29}$$

Recall the integral form of the log function:

$$\log(1+x) = \int_0^x \frac{1}{1+t} dt.$$

We integrate both sides of the Equality (29) and show that the numerator of $f_L(x)$ is equal to:

$$\log(1+x) - \sum_{k=1}^{K} \frac{(-1)^{k-1}}{k} x^k = (-1)^K \int_0^x \frac{t^K}{1+t} dt.$$

This result enables us to rewrite the function $f_L$ as:

$$f_L(x) = \frac{1}{x^{L+1}} \int_0^x \frac{t^L}{1+t} dt.$$

Using the change of variable $t = ux$, we obtain:

$$f_L(x) = \int_0^1 \frac{u^L}{1+xu} dt$$

which is clearly decreasing for in $\mathbb{R}^+$. This ends the proof. □

Finally, we also state the important change of measure lemma:

**Lemma 14** (Change of measure). *Let $g$ be a function of the parameter $\theta$ and data $\mathcal{D}_n$, for any distribution $Q$ that is $P$ continuous, for any $\delta \in (0,1]$, we have with probability $1 - \delta$ :*

$$\mathbb{E}_{\theta \sim Q}[g(\theta, \mathcal{D}_n)] \leq \mathcal{KL}(Q||P) + \ln \frac{\Psi_g}{\delta} \tag{30}$$

*with $\Psi_g = \mathbb{E}_{\mathcal{D}_n} \mathbb{E}_{\theta \sim P}[e^{g(\theta, \mathcal{D}_n)}]$.*

Lemma 14 is the backbone of a multitude of PAC-Bayesian bounds. It is proven in many references, see for example [2] or Lemma 1.1.3 in [11]. With this result, the recipe of constructing a generalization bound reduces to choosing an adequate function $g$ for which we can control $\Psi_g$.

# E    Additional results and discussions

## E.1    Plots of the empirical moments bounds (Proposition 1)

For any $\pi \in \Pi$, let $U_L^{\lambda, h}(\pi)$ be the upper bound of Proposition 1:

$$U_L^{\lambda, h}(\pi) = \psi_\lambda \left( \hat{R}_n^h(\pi) + \frac{\ln(1/\delta)}{\lambda n} + \sum_{\ell=2}^{2L} \frac{\lambda^{\ell-1}}{\ell} \hat{\mathcal{M}}_n^{h,\ell}(\pi) \right).$$

One can observe that the bound $U_L^{\lambda, h}$ depends on three parameters, the regularized IPS function $h$, the free parameter $\lambda$ and the moment order $L$. We choose a dataset (balance-scale) with $n = 612$, and evaluate a policy $\pi$ with $R(\pi) = -0.93$ to evaluate our bound for different parameters. We fix $\lambda = \sqrt{1/n}$ and plot the value of $U_L^{\lambda, h}$ for different values of the moment order $L \in \{1, 2, 3, 4, 6, 8, \infty\}$ and for 4 different regularization functions, namely IPS, clipped IPS ($M = \sqrt{n}$), Implicit Exploration (IX) ($\lambda = \sqrt{1/n}$) and Exponential Smoothing (ES) ($\alpha = 1 - \sqrt{1/n}$). The results are shown in Figure 3. One can observe from the plot that The decreasing nature of $U_L^{\lambda, h}$ depends on $\lambda$ and the regularization function $h$. Indeed, Proposition 2 states that $\lambda < \min_{i \in [n]} 1/|h_i|$ implies that the bound is decreasing w.r.t $L$. Which means that once this condition is not verified, we do not know if the bound will keep decreasing with $L$. If the bound seems decreasing for CIPS and IX, One can observe that for both IPS and ES, the bound increased from $L = 4$ to $L = 8$, but achieved its minimum at $L = \infty$, with IPS being optimal for this value. This highlights the connection between $L$, the value of $\lambda$ and the regularizer $h$.

## E.2    The study of Logarithmic Smoothing estimator and proofs

Recall the form of the Logarithmic Smoothing estimator, defined for any $\lambda \geq 0$:

$$\hat{R}_n^\lambda(\pi) = -\frac{1}{n} \sum_{i=1}^n \frac{1}{\lambda} \log \left( 1 - \lambda w_\pi(x_i, a_i) c_i \right). \tag{31}$$

Our estimator $\hat{R}_n^\lambda(\pi)$, is defined for a non-negative $\lambda \geq 0$. In particular, $\lambda = 0$ recovers the unbiased IPS estimator in (2) and $\lambda > 0$ introduces a bias variance trade-off. This estimator can be interpreted

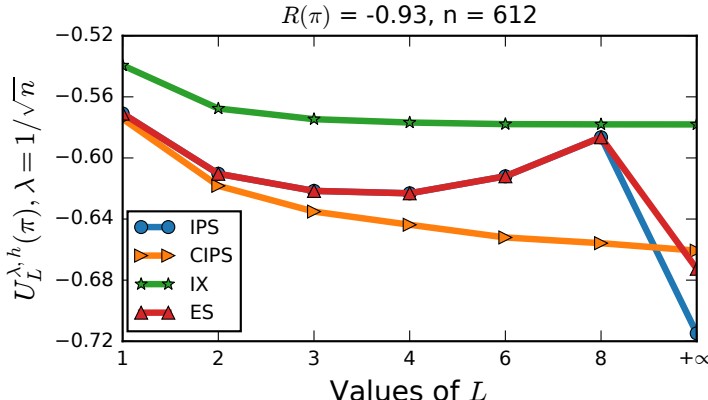

Figure 3: Proposition 1 for different values of $L$ and with different regularized IPS $h$.

as Logarithmic Soft Clipping, and have a similar behavior than Clipping of Bottou et al. [10]. Indeed, $1/\lambda$ plays a similar role to the clipping parameter $M$, as for any $i \in [n]$, we have:

$$w_\pi(x_i, a_i)c_i \ll \frac{1}{\lambda} \implies -\frac{1}{\lambda}\log(1 - \lambda w_\pi(x_i, a_i)c_i) \approx w_\pi(x_i, a_i)c_i.$$

$$w_\pi(x_i, a_i)c_i < M \implies \min(w_\pi(x_i, a_i), M)c_i = w_\pi(x_i, a_i)c_i.$$

LS can be seen as a smooth, differentiable version of clipping. We plot the graph of the two functions in Figure 4. One can observe that once $\lambda > 0$, LS exhibits a bias-variance trade-off, with a declining bias with $\lambda \to 0$. This is different than Clipping as no bias is suffered once $M$ is bigger than the support of $w_\pi$, this comes however with the price of suffering the full variance of IPS. In the following, we study the bias-variance trade-off that emerges with the new Logarithmic Smoothing estimator.

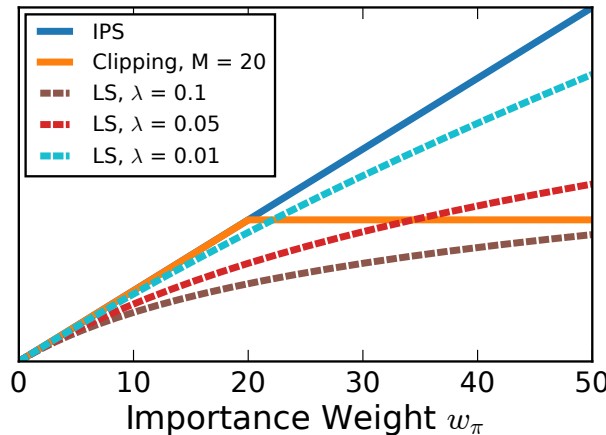

Figure 4: Comparison of Logarithmic Smoothing and Clipping.

We begin by defining the bias and variance of $\hat{R}_n^\lambda(\pi)$:

$$\mathcal{B}^\lambda(\pi) = \mathbb{E}\left[\hat{R}_n^\lambda(\pi)\right] - R(\pi), \qquad \mathcal{V}^\lambda(\pi) = \mathbb{E}\left[\left(\hat{R}_n^\lambda(\pi) - \mathbb{E}\left[\hat{R}_n^\lambda(\pi)\right]\right)^2\right]. \qquad (32)$$

Moreover, for any $\lambda \geq 0$, we define the following quantity

$$\mathcal{S}_\lambda(\pi) = \mathbb{E}\left[\frac{w_\pi(x, a)^2 c^2}{1 - \lambda w_\pi(x, a)c}\right], \qquad (33)$$

that will be essential in studying the properties of this estimator akin to the coverage ratio used for the IX-estimator [21]. In the following, we study the properties of our estimator $\hat{R}_n^\lambda(\pi)$ in (14). We start with bounding its mean squared error (MSE), which involves bounding its bias and variance.

> **Proposition** (Bias-variance trade-off). *Let $\pi \in \Pi$ and $\lambda \geq 0$. Then we have that*
> $$0 \leq \mathcal{B}^{\lambda}(\pi) \leq \lambda \mathcal{S}_{\lambda}(\pi), \quad \text{and} \quad \mathcal{V}^{\lambda}(\pi) \leq \mathcal{S}_{\lambda}(\pi)/n.$$
> *Moreover, it holds that for any $\lambda > 0$:*
> $$\mathcal{V}^{\lambda}(\pi) \leq \frac{|R(\pi)|}{n\lambda} \leq \frac{1}{n\lambda}.$$

*Proof.* Let us start with bounding the bias. We have for any $\lambda \geq 0$:

$$\mathcal{B}^{\lambda}(\pi) = \mathbb{E}\left[\hat{R}_n^{\lambda}(\pi)\right] - R(\pi)$$
$$= \mathbb{E}\left[-\frac{1}{\lambda}\log(1 - \lambda w_{\pi}(x,a)c) - w_{\pi}(x,a)c\right] \quad \text{(IPS is unbiased)}.$$

Using $\log(1+x) \leq x$ for any $x \geq 0$ proves that the bias is positive. For its upper bound, we use the following inequality $\log(1+x) \geq \frac{x}{1+x}$ holding for $x \geq 0$:

$$\mathcal{B}^{\lambda}(\pi) = \mathbb{E}\left[-\frac{1}{\lambda}\log(1 - \lambda w_{\pi}(x,a)c) - w_{\pi}(x,a)c\right]$$
$$\leq \mathbb{E}\left[\frac{w_{\pi}(x,a)c}{1 - \lambda w_{\pi}(x,a)c} - w_{\pi}(x,a)c\right] = \lambda\mathbb{E}\left[\frac{(w_{\pi}(x,a)c)^2}{1 - \lambda w_{\pi}(x,a)c}\right] = \lambda\mathcal{S}_{\lambda}(\pi).$$

Now focusing on the variance, we have:

$$\mathcal{V}^{\lambda}(\pi) = \mathbb{E}\left[\left(\hat{R}_n^{\lambda}(\pi) - \mathbb{E}\left[\hat{R}_n^{\lambda}(\pi)\right]\right)^2\right]$$
$$\leq \frac{1}{n\lambda^2}\mathbb{E}\left[\log(1 - \lambda w_{\pi}(x,a)c)^2\right].$$

We use the following inequality $\log(1+x) \leq x/\sqrt{x+1}$ holding for $x \geq 0$ to obtain our result:

$$\mathcal{V}^{\lambda}(\pi) \leq \frac{1}{n}\mathcal{S}_{\lambda}(\pi).$$

Notice that once $\lambda > 0$, we have:

$$\mathcal{S}_{\lambda}(\pi) = \mathbb{E}\left[\frac{w_{\pi}(x,a)^2 c^2}{1 - \lambda w_{\pi}(x,a)c}\right] \leq \frac{1}{\lambda}\mathbb{E}\left[w_{\pi}(x,a)|c|\right] = \frac{|R(\pi)|}{\lambda},$$

resulting in a finite variance whenever $\lambda > 0$:

$$\mathcal{V}^{\lambda}(\pi) \leq \frac{|R(\pi)|}{n\lambda} \leq \frac{1}{n\lambda}.$$

$\square$

$\lambda = 0$ recovers the IPS estimator in (2), with zero bias and variance bounded by $\mathbb{E}\left[w^2(x,a)c^2\right]/n$. When $\lambda > 0$, a bias-variance trade-off emerges. The bias is always non-negative as we still recover an estimator that verifies **(C1)**. The bias is capped at $\lambda\mathcal{S}_{\lambda}(\pi)$, which diminishes to zero when $\lambda$ is small and goes to $|R(\pi)|$ as $\lambda$ increases. Conversely, the variance decreases with a higher $\lambda$. Notably, $\lambda > 0$ ensures finite variance bounded by $1/\lambda n$, despite the estimator being unbounded. This is different from previous regularizations that relied on bounded functions to ensure finite variance.

While prior evaluations of estimators often relied on bias and variance analysis, Metelli et al. [38] argued for studying the non-asymptotic concentration rate of the estimators, advocating for sub-Gaussianity as a desired property. Even if our estimator is not bounded, we prove in the following that it is sub-Gaussian.

**Proposition** (Sub-Gaussianity). *Let $\pi \in \Pi$, $\delta \in (0,1]$ and $\lambda > 0$. Then the following inequalities holds with probability at least $1 - \delta$:*

$$R(\pi) - \hat{R}_n^\lambda(\pi) \leq \frac{\ln(2/\delta)}{\lambda n}, \qquad and \qquad \hat{R}_n^\lambda(\pi) - R(\pi) \leq \lambda \mathcal{S}_\lambda(\pi) + \frac{\ln(2/\delta)}{\lambda n}.$$

*In particular, setting $\lambda = \lambda_* = \sqrt{\ln(2/\delta)/n\mathbb{E}\left[w_\pi(x,a)^2 c^2\right]}$ yields that*

$$|R(\pi) - \hat{R}_n^{\lambda_*}(\pi)| \leq \sqrt{2\sigma^2 \ln(2/\delta)}, \qquad where\ \sigma^2 = 2\mathbb{E}\left[w_\pi(x,a)^2 c^2\right]/n. \qquad (34)$$

*Proof.* Let $\pi \in \Pi$, $\lambda > 0$ and $\delta > 0$. To prove sub-Gaussianity, we need both upper bounds and lower bounds on $R(\pi)$ using $\hat{R}_n^\lambda(\pi)$. For the upper bound, we can use the bound of Corollary 5, and recall that $\psi_\lambda(x) \leq x$ for all $x$. We then obtain with a probability $1 - \delta$:

$$R(\pi) \leq \psi_\lambda\left(\hat{R}_n^\lambda(\pi) + \frac{\ln(1/\delta)}{\lambda n}\right) \implies R(\pi) - \hat{R}_n^\lambda(\pi) \leq \frac{\ln(1/\delta)}{\lambda n}.$$

For the lower bound on the risk, we go back to our Chernoff Lemma 12, and use the collection of i.i.d. random variable, that for any $i \in [n]$, are defined as:

$$\bar{X}_i = -\frac{1}{\lambda}\log\left(1 - \lambda w_\pi(x_i, a_i) c_i\right).$$

This gives for $a \in \mathbb{R}$:

$$P\left(\sum_{i\in[n]} \bar{X}_i > a\right) \leq \left(\mathbb{E}\left[\exp\left(\lambda \bar{X}_1\right)\right]\right)^n \exp(-\lambda a)$$

$$P\left(\sum_{i\in[n]} \bar{X}_i > a\right) \leq \left(\mathbb{E}\left[\frac{1}{1 - \lambda w_\pi(x,a)c}\right]\right)^n \exp(-\lambda a)$$

Solving for $\delta = \left(\mathbb{E}\left[\frac{1}{1-\lambda w_\pi(x,a)c}\right]\right)^n \exp(-\lambda a)$, we get:

$$P\left(\frac{1}{n}\sum_{i\in[n]} \bar{X}_i > \frac{1}{\lambda}\log\left(\mathbb{E}\left[\frac{1}{1 - \lambda w_\pi(x,a)c}\right]\right) + \frac{\ln(1/\delta)}{\lambda n}\right) \leq \delta$$

The complementary event holds with at least probability $1 - \delta$:

$$\hat{R}_n^\lambda(\pi) \leq \frac{1}{\lambda}\log\left(\mathbb{E}\left[\frac{1}{1 - \lambda w_\pi(x,a)c}\right]\right) + \frac{\ln(1/\delta)}{\lambda n},$$

which implies using the inequality $\log(x) \leq x - 1$ for all $x > 0$:

$$\hat{R}_n^\lambda(\pi) - R(\pi) \leq \frac{1}{\lambda}\log\left(\mathbb{E}\left[\frac{1}{1 - \lambda w_\pi(x,a)c}\right]\right) - R(\pi) + \frac{\ln(1/\delta)}{\lambda n}$$

$$\leq \frac{1}{\lambda}\left(\mathbb{E}\left[\frac{1}{1 - \lambda w_\pi(x,a)c}\right] - 1\right) - R(\pi) + \frac{\ln(1/\delta)}{\lambda n}$$

$$\leq \mathbb{E}\left[\frac{w_\pi(x,a)c}{1 - \lambda w_\pi(x,a)c} - w_\pi(x,a)c\right] + \frac{\ln(1/\delta)}{\lambda n}$$

$$\leq \lambda\mathbb{E}\left[\frac{w_\pi(x,a)^2 c^2}{1 - \lambda w_\pi(x,a)c}\right] + \frac{\ln(1/\delta)}{\lambda n} = \lambda\mathcal{S}_\lambda(\pi) + \frac{\ln(1/\delta)}{\lambda n},$$

which proves the lower bound on the risk. As both results hold with high probability, we use a union argument to have them both holding for probability at least $1 - \delta$:

$$R(\pi) - \hat{R}_n^\lambda(\pi) \leq \frac{\ln(2/\delta)}{\lambda n}, \qquad and \qquad \hat{R}_n^\lambda(\pi) - R(\pi) \leq \lambda\mathcal{S}_\lambda(\pi) + \frac{\ln(2/\delta)}{\lambda n},$$

which implies that:

$$|R(\pi) - \hat{R}_n^\lambda(\pi)| \leq \lambda \mathcal{S}_\lambda(\pi) + \frac{\ln(2/\delta)}{\lambda n} \leq \lambda \mathbb{E}\left[w_\pi(x,a)^2 c^2\right] + \frac{\ln(2/\delta)}{\lambda n}.$$

This means that setting $\lambda = \lambda_* = \sqrt{\ln(2/\delta)/n\mathbb{E}\left[w_\pi(x,a)^2 c^2\right]}$ yields a sub-Gaussian concentration:

$$|R(\pi) - \hat{R}_n^{\lambda_*}(\pi)| \leq 2\sqrt{\frac{\mathbb{E}\left[w_\pi(x,a)^2 c^2\right]\ln(2/\delta)}{n}}.$$

This ends the proof. $\qquad\qquad\square$

From (34), $\hat{R}_n^{\lambda_*}(\pi)$ is sub-Gaussian with variance proxy $\sigma^2 = 2\mathbb{E}\left[\omega(x,a)^2 c^2\right]/n$, which is lower that the variance proxy of the Harmonic estimator of Metelli et al. [38]. Indeed, the Harmonic estimator has a slightly worse variance proxy of $\sigma_H^2 = \frac{(2+\sqrt{3})^2}{3}\mathbb{E}\left[\omega(x,a)^2 c^2\right]/n$, giving $\sigma^2 < \sigma_H^2$.

### E.3   OPS: Formal comparison with IX suboptimality

Let us begin by stating results from the IX work [21]. Recall that the IX estimator is defined for any $\lambda > 0$, by:

$$\hat{R}_n^{\lambda\text{-IX}}(\pi) = \frac{1}{n}\sum_{i=1}^n \frac{\pi(a_i|x_i)}{\pi_0(a_i|x_i) + \lambda/2}c_i.$$

Let $\Pi_s = \{\pi_1, ..., \pi_m\}$ be a finite set of predefined policies. In OPS, the goal is to find $\pi_*^s \in \Pi_s$ that satisfies

$$\pi_*^s = \operatorname{argmin}_{\pi\in\Pi_s} R(\pi) = \operatorname{argmin}_{k\in[m]} R(\pi_k).$$

for $\lambda > 0$, the selection strategy suggested in Gabbianelli et al. [21] was to search for:

$$\hat{\pi}_n^{s,\text{ IX}} = \operatorname*{argmin}_{\pi\in\Pi_s} \hat{R}_n^{\lambda\text{-IX}}(\pi) = \operatorname*{argmin}_{k\in[m]} \hat{R}_n^{\lambda\text{-IX}}(\pi). \qquad (35)$$

---

**Proposition 15** (Suboptimality of the IX selection strategy). *Let $\lambda > 0$ and $\delta \in (0,1]$. Then, it holds with probability at least $1 - \delta$ that*

$$0 \leq R(\hat{\pi}_n^{s,\text{ IX}}) - R(\pi_*^s) \leq \lambda\mathcal{C}_{\lambda/2}(\pi_*^s) + \frac{2\ln(2|\Pi_s|/\delta)}{\lambda n}, \qquad (36)$$

*where*

$$\mathcal{C}_\lambda(\pi) = \mathbb{E}\left[\frac{\pi(a|x)}{\pi_0^2(a|x) + \lambda\pi_0(a|x)}|c|\right].$$

---

Both suboptimalities (LS and IX) have the same form, they only depend on two different quantities ($\mathcal{S}_\lambda$ and $\mathcal{C}_\lambda$ respectively). For a $\pi \in \Pi$ and $\lambda > 0$, If we can identify when $\mathcal{S}_\lambda(\pi) \leq \mathcal{C}_{\lambda/2}(\pi)$, then we can prove that the sub-optimality of LS selection strategy is better than the one of IX. Luckily, this is always the case, and it is stated formally below.

---

**Proposition 16.** *Let $\pi \in \Pi$ and $\lambda > 0$. We have:*

$$\mathcal{S}_\lambda(\pi) \leq \mathcal{C}_{\lambda/2}(\pi). \qquad (37)$$

---

*Proof.* Let $\pi \in \Pi$ and $\lambda > 0$, we have:

$$\mathcal{C}_{\lambda/2}(\pi) - \mathcal{S}_\lambda(\pi) = \mathbb{E}\left[\frac{\pi(a|x)}{\pi_0^2(a|x) + \frac{\lambda}{2}\pi_0(a|x)}|c| - \frac{w_\pi(x,a)^2 c^2}{1 - \lambda w_\pi(x,a)c}\right]$$

$$= \mathbb{E}\left[\frac{\pi(a|x)}{\pi_0^2(a|x) + \frac{\lambda}{2}\pi_0(a|x)}|c| - \frac{\pi(a|x)^2 c^2}{\pi_0^2(a|x) - \lambda\pi_0(a|x)\pi(a|x)c}\right]$$

$$= \mathbb{E}\left[\pi(a|x)|c|\left(\frac{1}{\pi_0^2(a|x) + \frac{\lambda}{2}\pi_0(a|x)} - \frac{\pi(a|x)|c|}{\pi_0^2(a|x) + \lambda\pi_0(a|x)\pi(a|x)|c|}\right)\right]$$

$$= \mathbb{E}\left[\pi(a|x)|c|\left(\frac{\pi_0^2(a|x)\left(1 - \pi(a|x)|c|\right) + \frac{\lambda}{2}\pi_0(a|x)\pi(a|x)|c|}{(\pi_0^2(a|x) + \frac{\lambda}{2}\pi_0(a|x))(\pi_0^2(a|x) + \lambda\pi_0(a|x)\pi(a|x)|c|)}\right)\right]$$

$$\geq 0.$$

$\square$

This means that the suboptimality of LS selection strategy is better bounded than the one of IX. Our experiments confirm that the LS selection strategy is better than IX in practical scenarios.

**Minimax optimality of our selection strategy.** As discussed in Gabbianelli et al. [21], pessimistic algorithms tend to have the property that their regret scales with the minimax sample complexity of estimating the value of the optimal policy [27]. For the case of multi-armed bandit (one context $x$), this estimation minimax sample complexity is proved by Li et al. [34] and is of the rate $\mathcal{O}(\mathbb{E}[w_{\pi^*}(x,a)^2 c^2])$, with $\pi^*$ being the optimal policy. Our bound matches the lower bound proved by Li et al. [34], as:

$$\mathcal{S}_\lambda(\pi^*) = \mathbb{E}\left[\frac{w_{\pi^*}(x,a)^2 c^2}{1 - \lambda w_{\pi^*}(x,a)c}\right] \leq \mathbb{E}\left[w_{\pi^*}(x,a)^2 c^2\right],$$

which is not the case for the suboptimality of IX, that only matches it in the deterministic setting with binary costs, as:

$$\mathcal{C}_\lambda(\pi^*) = \mathbb{E}\left[\frac{\pi^*(a|x)}{\pi_0^2(a|x) + \lambda\pi_0(a|x)}|c|\right] \leq \mathbb{E}\left[\frac{\pi^*(a|x)}{\pi_0^2(a|x)}|c|\right] = \mathbb{E}\left[\left(\frac{\pi^*(a|x)}{\pi_0(a|x)}\right)^2 c^2\right],$$

with the last inequality only holding when $\pi^*$ is deterministic and the costs are binary. For deterministic policies and the general contextual bandit, we invite the reader to see a formal proof of the minimax lower bound of pessimism in Jin et al. [28, Theorem 4.4], matched for both IX and LS.

### E.4 OPL: Formal comparison of PAC-Bayesian bounds

As it is easier to work with linear estimators within the PAC-Bayesian framework, we define the following estimator of the risk $\hat{R}_n^{p-\text{LIN}}(\pi)$, with the help of a function $p : \mathbb{R} \to \mathbb{R}$ as:

$$\hat{R}_n^{p-\text{LIN}}(\pi) = \frac{1}{n}\sum_{i=1}^n \frac{\pi(a_i|x_i)}{p(\pi_0(a_i|x_i))}c_i$$

with the only condition on $p$ to be $\{C_1^{\text{LIN}} : \forall x, p(x) \geq x\}$. This condition helps us control the impact of actions with low probabilities under $\pi_0$. This risk estimator encompasses well known risk estimators depending on the choice of $p$.

Now that we defined the family of estimators covered by our analysis, we attack the problem of deriving generalization bounds. We derive our empirical high order bound expressed in the following:

**Proposition 17** (Empirical High Order PAC-Bayes bound). *Let $L \geq 1$. Given a prior $P$ on $\mathcal{F}_\Theta$, $\delta \in (0,1]$ and $\lambda > 0$, the following bound holds with probability at least $1 - \delta$ uniformly for all distribution $Q$ over $\mathcal{F}_\Theta$:*

$$R(\pi_Q) \leq \psi_\lambda \left( \hat{R}_n^{p-\text{LIN}}(\pi_Q) + \frac{\mathcal{KL}(Q||P) + \ln \frac{1}{\delta}}{\lambda n} + \sum_{\ell=2}^{2L} \frac{\lambda^{\ell-1}}{\ell} \hat{\mathcal{M}}_n^{p-\text{LIN},\ell}(\pi_Q) \right) \quad (38)$$

*with:*

$$\hat{\mathcal{M}}_n^{p-\text{LIN},\ell}(\pi_Q) = \frac{1}{n} \sum_{i=1}^n \frac{\pi_Q(a_i|x_i)}{p(\pi_0(a_i|x_i))^\ell} c_i^\ell$$

$$\psi_\lambda = x :\to \frac{1 - \exp(-\lambda x)}{\lambda}.$$

*Proof.* Let $L \geq 1$, we have from Lemma 13, and for any positive random variable $X \geq 0$ and $\lambda > 0$:

$$f_{2L-1}(0) = \frac{1}{2L} \geq f_{2L-1}(\lambda X) = -\frac{\log(1 + \lambda X) - \sum_{\ell=1}^{2L-1} \frac{(-1)^{\ell-1}}{\ell}(\lambda X)^\ell}{(\lambda X)^{2L}}$$

which is equivalent to:

$$\sum_{\ell=1}^{2L} \frac{(-1)^{\ell-1}}{\ell}(\lambda X)^\ell \leq \log(1 + \lambda X) \iff \exp\left( \sum_{\ell=1}^{2L} \frac{(-1)^{\ell-1}}{\ell}(\lambda X)^\ell \right) \leq 1 + \lambda X$$

$$\implies \mathbb{E}\left[ \exp\left( \sum_{\ell=1}^{2L} \frac{(-1)^{\ell-1}}{\ell}(\lambda X)^\ell \right) \right] \leq 1 + \mathbb{E}\left[\lambda X\right]$$

$$\implies \mathbb{E}\left[ \exp\left( \sum_{\ell=1}^{2L} \frac{(-1)^{\ell-1}}{\ell}(\lambda X)^\ell \right) \right] \leq \exp\left( \log(1 + \mathbb{E}\left[\lambda X\right]) \right),$$

which implies that:

$$\mathbb{E}\left[ \exp\left( \lambda(X - \frac{1}{\lambda}\log(1 + \mathbb{E}\left[\lambda X\right])) + \sum_{\ell=2}^{2L} \frac{(-1)^{\ell-1}}{\ell}(\lambda X)^\ell \right) \right] \leq 1.$$

For any $X \leq 0$, we can inject $-X \geq 0$ to obtain:

$$\forall X \leq 0, \quad \mathbb{E}\left[ \exp\left( \lambda\left( -\frac{1}{\lambda}\log(1 + \mathbb{E}\left[\lambda X\right]) - X \right) - \sum_{k=2}^{2K} \frac{1}{k}(\lambda X)^k \right) \right] \leq 1. \quad (39)$$

Let:

$$d_\theta(a|x) = \mathbb{1}\left[ f_\theta(x) = a \right], \forall(x,a) \in \mathcal{X} \times \mathcal{A},$$

it means that:

$$\pi_Q(a|x) = \mathbb{E}_{\theta \sim Q}\left[ d_\theta(a|x) \right], \forall(x,a) \in \mathcal{X} \times \mathcal{A}.$$

Let $\lambda > 0$. The adequate function $g$ we are going to use in combination with Lemma 14 is:

$$g(\theta, \mathcal{D}_n) = \sum_{i=1}^n \lambda \left( -\frac{1}{\lambda}\log(1 + \lambda R^{p-\text{LIN}}(d_\theta)) - \frac{d_\theta(a_i|x_i)}{p(\pi_0(a_i|x_i))}c_i \right) - \sum_{\ell=2}^{2L} \frac{1}{\ell} \left( \lambda \frac{d_\theta(a_i|x_i)}{p(\pi_0(a_i|x_i))}c_i \right)^\ell$$

$$= \sum_{i=1}^n \lambda \left( -\frac{1}{\lambda}\log(1 + \lambda R^{p-\text{LIN}}(d_\theta)) - \frac{d_\theta(a_i|x_i)}{p(\pi_0(a_i|x_i))}c_i \right) - \sum_{\ell=2}^{2L} \frac{d_\theta(a_i|x_i)}{\ell} \left( \frac{\lambda}{p(\pi_0(a_i|x_i))}c_i \right)^\ell.$$

By exploiting the i.i.d. nature of the data and exchanging the order of expectations ($P$ is independent of $\mathcal{D}_n$), we can naturally prove using (39) that:

$$\Psi_g = \mathbb{E}_P \left[ \prod_{i=1}^n \mathbb{E} \left[ \exp \left( \lambda \left( -\frac{1}{\lambda} \log(1 + \lambda R^{p-\text{LIN}}(d_\theta)) - X_i(\theta) \right) - \sum_{k=2}^{2K} \frac{1}{k} \left( \lambda X_i(\theta) \right)^k \right) \right] \right] \leq 1,$$

as we have :

$$X_i(\theta) = \frac{d_\theta(a_i|x_i)}{p(\pi_0(a_i|x_i))} c_i \leq 0 \quad \forall i.$$

Injecting $\Psi_g$ in Lemma 14, rearranging terms and using that $\hat{R}_n^{p-\text{LIN}}(\pi)$ has positive bias concludes the proof. $\qquad\square$

Similarly to the OPE section, we use this general bound to obtain a PAC-Bayesian Empirical Second Moment bound and the PAC-Bayesian LS-LIN bound. That we state directly below:

**Empirical second moment bound.** With $L = 1$, we obtain the following:

> **Corollary 18** (Second Moment Upper bound). *Given a prior $P$ on $\mathcal{F}_\Theta$, $\delta \in (0, 1]$ and $\lambda > 0$. The following bound holds with probability at least $1 - \delta$ uniformly for all distribution $Q$ over $\mathcal{F}_\Theta$:*
>
> $$R(\pi_Q) \leq \psi_\lambda \left( \hat{R}_n^p(\pi_Q) + \frac{\mathcal{KL}(Q||P) + \ln\frac{1}{\delta}}{\lambda n} + \frac{\lambda}{2} \hat{\mathcal{M}}_n^{p-\text{LIN},2}(\pi_Q) \right). \tag{40}$$

**Log Smoothing PAC-Bayesian Bound.** With $L \to \infty$, we obtain the following:

> **Proposition 19** ($\hat{R}_n^{\lambda-\text{LIN}}$ PAC-Bayes bound). *Given a prior $P$ on $\mathcal{F}_\Theta$, $\delta \in (0, 1]$ and $\lambda > 0$, the following bound holds with probability at least $1 - \delta$ uniformly for all distribution $Q$ over $\mathcal{F}_\Theta$:*
>
> $$R(\pi_Q) \leq \psi_\lambda \left( \hat{R}_n^{\lambda-\text{LIN}}(\pi_Q) + \frac{\mathcal{KL}(Q||P) + \ln\frac{1}{\delta}}{\lambda n} \right). \tag{41}$$
>
> *with:*
>
> $$\hat{R}_n^{\lambda\text{-LIN}}(\pi) = -\frac{1}{n} \sum_{i=1}^n \frac{\pi(a_i|x_i)}{\lambda} \log \left( 1 - \frac{\lambda c_i}{\pi_0(a_i|x_i)} \right).$$

Following the same proof schema as of the OPE section, we can demonstrate that the Log Smoothing PAC-Bayesian bound dominates the Empirical Second moment PAC-Bayesian bound $L = 1$. However, we use the bound of $L = 1$ as an intermediary to state the dominance of the Log Smoothing PAC-Bayesian bound.

Indeed, we can easily compare the result obtained with $L = 1$ to previously derived PAC-Bayesian bounds for off-policy learning. We start by writing down the conditional Bernstein bound of Sakhi et al. [49] holding for the (linear) cIPS ($p : x \to \max(x, \tau)$). For a policy $\pi_Q$ and a $\lambda > 0$, we have:

$$R(\pi_Q) \leq \hat{R}_n^\tau(\pi_Q) + \sqrt{\frac{\mathcal{KL}(Q||P) + \ln\frac{4\sqrt{n}}{\delta}}{2n}} + \frac{\mathcal{KL}(Q||P) + \ln\frac{2}{\delta}}{\lambda n} + \lambda g\left(\lambda/\tau\right) \mathcal{V}_n^\tau(\pi_Q).$$

$$R(\pi_Q) \leq \hat{R}_n^\tau(\pi_Q) + \frac{\mathcal{KL}(Q||P) + \ln\frac{1}{\delta}}{\lambda n} + \frac{\lambda}{2} \hat{\mathcal{S}}_n^\tau(\pi_Q). \tag{L = 1}$$

We can observe that the previously derived conditional Bernstein bound has several terms that make it less tight:

- It has an additional, strictly positive square root KL divergence term.

- The multiplicative factor $g(\lambda/\tau)$ is always bigger than $1/2$, and diverges when $\tau \to 0$.
- With enough data ($n \gg 1$), we also have:

$$\hat{S}_n^\tau(\pi_Q) \approx \mathbb{E}\left[\frac{\pi_Q(a|x)}{\max\{\pi_0(a|x), \tau\}^2}c(a,x)^2\right] \leq \mathbb{E}\left[\frac{\pi_Q(a|x)}{\max\{\pi_0(a|x), \tau\}^2}\right] \approx \mathcal{V}_n^\tau(\pi_Q).$$

These observations confirm that the new bound derived with $L = 1$ is tighter than what was previously proposed for `cIPS`, especially when $n \gg 1$. As our bound can work for other estimators, we also compare it to a recently proposed PAC-Bayes bound in Aouali et al. [5] for the exponentially-smoothed estimator ($p : x \to x^\alpha$) with $\alpha \in [0,1]$:

$$R(\pi_Q) \leq \hat{R}_n^\alpha(\pi_Q) + \sqrt{\frac{\mathcal{KL}(Q||P) + \ln\frac{4\sqrt{n}}{\delta}}{2n}} + \frac{\mathcal{KL}(Q||P) + \ln\frac{2}{\delta}}{\lambda n} + \frac{\lambda}{2}\left(\mathcal{V}_n^\alpha(\pi_Q) + \hat{S}_n^\alpha(\pi_Q)\right).$$

$$R(\pi_Q) \leq \hat{R}_n^\alpha(\pi_Q) + \frac{\mathcal{KL}(Q||P) + \ln\frac{1}{\delta}}{\lambda n} + \frac{\lambda}{2}\hat{S}_n^\alpha(\pi_Q). \tag{L = 1}$$

We can clearly see that the previously proposed bound for the exponentially smoothed estimator has two additional positive quantities that makes it less tight than our bound. In addition, computing our bound does not rely on expectations under $\pi_0$ (contrary to the previous bounds that have $\mathcal{V}_n$) which alleviates the need to access the logging policy and reduce the computations.

This demonstrates the superiority of $L = 1$ compared to existing variance sensitive PAC-Bayesian bounds. It means that $L \to \infty$ is even better. We can also prove that the Log smoothing PAC-Bayesian Bound is better than the one of IX in Gabbianelli et al. [21]. Indeed, using $\log(1 + x) \geq \frac{x}{1+x/2}$ for all $x \geq 0$, we have for any $P, Q \in \mathcal{P}(\Theta)$ and $\lambda > 0$:

$$
\begin{aligned}
\psi_\lambda\left(\hat{R}_n^{\lambda-\text{LIN}}(\pi_Q) + \frac{\mathcal{KL}(Q||P) + \ln\frac{1}{\delta}}{\lambda n}\right) &\leq \hat{R}_n^{\lambda-\text{LIN}}(\pi_Q) + \frac{\mathcal{KL}(Q||P) + \ln\frac{1}{\delta}}{\lambda n} \\
&\leq -\frac{1}{n}\sum_{i=1}^n \frac{\pi_Q(a_i|x_i)}{\lambda}\log\left(1 - \frac{\lambda c_i}{\pi_0(a_i|x_i)}\right) + \frac{\mathcal{KL}(Q||P) + \ln\frac{1}{\delta}}{\lambda n} \\
&\leq \frac{1}{n}\sum_{i=1}^n \frac{\pi_Q(a_i|x_i)}{\pi_0(a_i|x_i) - \lambda c_i/2} + \frac{\mathcal{KL}(Q||P) + \ln\frac{1}{\delta}}{\lambda n} \\
&\leq \hat{R}_n^{\lambda-\text{IX}}(\pi_Q) + \frac{\mathcal{KL}(Q||P) + \ln\frac{1}{\delta}}{\lambda n}, \quad \textbf{(IX-bound)}
\end{aligned}
$$

with the last implication leveraging that the cost is always bigger than $-1/$ This proves that our bound is better than the IX bound. This means that our PAC-Bayesian bound is better than all existing PAC-Bayesian off-policy learning bounds.

### E.5  OPL: Formal comparison with IX PAC-Bayesian learning suboptimality

Let us begin by stating results from the IX work [21]. Recall that the IX estimator is defined for any $\lambda > 0$, by:

$$\hat{R}_n^{\text{IX}-\lambda}(\pi) = \frac{1}{n}\sum_{i=1}^n \frac{\pi(a_i|x_i)}{\pi_0(a_i|x_i) + \lambda/2}c_i,$$

and that we used the linearized version of the `LS` estimator, `LS-LIN` defined as:

$$\hat{R}_n^{\lambda\text{-LIN}}(\pi) = -\frac{1}{n}\sum_{i=1}^n \frac{\pi(a_i|x_i)}{\lambda}\log\left(1 - \frac{\lambda c_i}{\pi_0(a_i|x_i)}\right).$$

Let $\Theta$ be a parameter space and $\mathcal{P}(\Theta)$ be the set of all probability distribution on $\Theta$. Our goal is to find the best policy in a chosen class $\mathcal{L}(\Theta) \subset \mathcal{P}(\Theta)$:

$$\pi_{Q^*} = \underset{Q \in \mathcal{L}(\Theta)}{\text{argmin}}\, R(\pi_Q).$$

For $\lambda > 0$ and a prior $P \in \mathcal{P}(\Theta)$, the PAC-Bayesian learning strategy suggested in Gabbianelli et al. [21] is to find in $\mathcal{L}(\Theta) \subset \mathcal{P}(\Theta)$:

$$\hat{\pi}_{Q_n}^{\mathrm{IX}} = \underset{Q \in \mathcal{L}(\Theta)}{\operatorname{argmin}} \left\{ \hat{R}_n^{\mathrm{IX}-\lambda}(\pi_Q) + \frac{\mathcal{KL}(Q||P)}{\lambda n} \right\}.$$

This learning strategy suffers from a suboptimality bounded in the result below:

---

**Proposition 20** (Suboptimality of the IX PAC-Bayesian learning strategy from [21]). *Let $\lambda > 0$ and $\delta \in (0, 1]$. Then, it holds with probability at least $1 - \delta$ that*

$$0 \le R(\hat{\pi}_{Q_n}^{\mathrm{IX}}) - R(\pi_{Q^*}) \le \lambda \mathcal{C}_{\lambda/2}(\pi_{Q^*}) + \frac{2\left(\mathcal{KL}(Q^*||P) + \ln(2/\delta)\right)}{\lambda n},$$

*where*

$$\mathcal{C}_\lambda(\pi) = \mathbb{E}\left[ \frac{\pi(a|x)}{\pi_0^2(a|x) + \lambda \pi_0(a|x)} |c| \right].$$

---

Similarly for PAC-Bayesian learning, both suboptimalities (LS and IX) have the same form, they only depend on two different quantities ($\mathcal{S}_\lambda^{\mathrm{LIN}}$ and $\mathcal{C}_\lambda$ respectively). For a $\pi \in \Pi$ and $\lambda > 0$, If we can identify when $\mathcal{S}_\lambda^{\mathrm{LIN}}(\pi) \le \mathcal{C}_{\lambda/2}(\pi)$, then we can prove that the sub-optimality of LS PAC-Bayesian learning strategy is better than the one of IX in certain cases. Luckily, this is always the case, and it is stated formally below.

---

**Proposition 21.** *Let $\pi \in \Pi$ and $\lambda > 0$. We have:*

$$\mathcal{S}_\lambda^{\mathrm{LIN}}(\pi) \le \mathcal{C}_{\lambda/2}(\pi). \tag{42}$$

---

*Proof.* Let $\pi \in \Pi$ and $\lambda > 0$, and recall that:

$$\mathcal{S}_\lambda^{\mathrm{LIN}}(\pi) = \mathbb{E}\left[ \frac{\pi(a|x)c^2}{\pi_0^2(a|x) - \lambda \pi_0(a|x)c} \right].$$

We have:

$$\mathcal{C}_{\lambda/2}(\pi) - \mathcal{S}_\lambda^{\mathrm{LIN}}(\pi) = \mathbb{E}\left[ \frac{\pi(a|x)}{\pi_0^2(a|x) + \frac{\lambda}{2}\pi_0(a|x)}|c| - \frac{\pi(a|x)c^2}{\pi_0^2(a|x) - \lambda \pi_0(a|x)c} \right]$$

$$= \mathbb{E}\left[ \pi(a|x)|c| \left( \frac{1}{\pi_0^2(a|x) + \frac{\lambda}{2}\pi_0(a|x)} - \frac{|c|}{\pi_0^2(a|x) + \lambda \pi_0(a|x)|c|} \right) \right]$$

$$= \mathbb{E}\left[ \pi(a|x)|c| \left( \frac{\pi_0^2(a|x)(1 - |c|) + \frac{\lambda}{2}\pi_0(a|x)|c|}{(\pi_0^2(a|x) + \frac{\lambda}{2}\pi_0(a|x))(\pi_0^2(a|x) + \lambda \pi_0(a|x)|c|)} \right) \right]$$

$$\ge 0.$$

$\square$

Similarly, this means that the suboptimality of LS-LIN PAC-Bayesian learning strategy is also, better bounded than the one of IX.

**Minimax optimality of our learning strategy.** From Jin et al. [28, Theorem 4.4] we can state that the minimax suboptimality lower bound, in the case of deterministic optimal policies is of the rate $\mathcal{O}(1/\sqrt{nC^*})$ with $\inf_{x \in \mathcal{X}} \pi_0(\pi^*(x)|x) > C^*$. Our bound as well as IX bound match this minimax lower bound, as:

$$\mathcal{S}_\lambda^{\mathrm{LIN}}(\pi^*) = \mathbb{E}_{x,c}\left[ \frac{c^2}{\pi_0(\pi^*(x)|x) - \lambda c} \right] \le \frac{1}{C^*}$$

$$\mathcal{C}_\lambda(\pi^*) = \mathbb{E}_{x,c}\left[ \frac{|c|}{\pi_0(\pi^*(x)|x) + \lambda} \right] \le \frac{1}{C^*}.$$

One can see that for both, selecting a

$$\lambda^* = \sqrt{\frac{2\left(\mathcal{KL}(Q^*||P) + \ln(2/\delta)\right)C^*}{n}},$$

gets you the desired bound, matching this minimax rate.

# F   Proofs of OPE

## F.1   Proof of high order empirical moments bound (Proposition 1)

**Proposition** (Empirical moments risk bound). *Let* $\pi \in \Pi$, $L \geq 1$, $\delta \in (0,1]$, $\lambda > 0$ *and* $h$ *satisfying* (**C1**). *Then it holds with probability at least* $1 - \delta$ *that*

$$R(\pi) \leq \psi_\lambda\left(\hat{R}_n^h(\pi) + \sum_{\ell=2}^{2L} \frac{\lambda^{\ell-1}}{\ell}\hat{\mathcal{M}}_n^{h,\ell}(\pi) + \frac{\ln(1/\delta)}{\lambda n}\right),$$

*where* $\psi_\lambda$ *and* $\hat{\mathcal{M}}_n^{h,\ell}(\pi)$ *are defined in* (5), *respectively, and recall that* $\psi_\lambda(x) \leq x$.

*Proof.* Let $L \in \mathbb{N}^*$, $\lambda > 0$ and $X \geq 0$ a **positive random variable**. We have $2L - 1 \geq 1$, and with the decreasing nature of $f_{(2L-1)}$ (Lemma 13), we also have:

$$f_{(2L-1)}(0) \geq f_{2L-1}(\lambda X) \iff \frac{1}{2L} \geq -\frac{\log(1+\lambda X) - \sum_{l=1}^{2L-1}\frac{(-1)^{\ell-1}}{k}(\lambda X)^\ell}{(\lambda X)^{2L}}$$

$$\iff \sum_{\ell=1}^{2L}\frac{(-1)^{\ell-1}}{k}(\lambda X)^\ell \leq \log(1+\lambda X)$$

$$\iff \exp\left(\sum_{\ell=1}^{2L}\frac{(-1)^{\ell-1}}{\ell}(\lambda X)^\ell\right) \leq 1+\lambda X$$

$$\implies \mathbb{E}\left[\exp\left(\sum_{\ell=1}^{2L}\frac{(-1)^{\ell-1}}{\ell}(\lambda X)^\ell\right)\right] \leq 1+\lambda\mathbb{E}[X]$$

$$\implies \mathbb{E}\left[\exp\left(\sum_{\ell=1}^{2L}\frac{(-1)^{\ell-1}}{\ell}(\lambda X)^\ell\right)\right] \leq \exp\left((\log(1+\lambda\mathbb{E}[X]))\right)$$

$$\implies \mathbb{E}\left[\exp\left(\lambda(X - \frac{1}{\lambda}\log\left(1+\lambda\mathbb{E}[X]\right)) + \sum_{\ell=2}^{2L}\frac{(-1)^{\ell-1}}{\ell}(\lambda X)^\ell\right)\right] \leq 1.$$

For any $X \leq 0$, we can inject $-X \geq 0$ to obtain:

$$\forall X \leq 0, \quad \mathbb{E}\left[\exp\left(\lambda\left(-\frac{1}{\lambda}\log\left(1-\lambda\mathbb{E}[X]\right) - X\right) - \sum_{\ell=2}^{2L}\frac{1}{\ell}(\lambda X)^\ell\right)\right] \leq 1. \qquad (43)$$

The result in Equation (43) will be combined with Chernoff Inequality (Lemma 12) to finally prove our bound. Let $\lambda > 0$, for our problem, we define the random variable $X_i$ to use in the Chernoff Inequality as:

$$X_i = -\frac{1}{\lambda}\log\left(1-\lambda\mathbb{E}[h]\right) - h_i - \sum_{\ell=2}^{2L}\frac{1}{\ell}(\lambda h_i)^\ell.$$

For any $a \in \mathbb{R}$, this gives us the following:

$$P\left(\sum_{i \in [n]} X_i > a\right) \leq \left(\mathbb{E}\left[\exp\left(\lambda X_1\right)\right]\right)^n \exp(-\lambda a)$$

$$P\left(-\frac{n}{\lambda}\log\left(1 - \lambda \mathbb{E}\left[h\right]\right) - \sum_{i \in [n]}\left(h_i + \sum_{\ell=2}^{2L}\frac{1}{\ell}(\lambda h_i)^\ell\right) > a\right) \leq \left(\mathbb{E}\left[\exp\left(\lambda X_1\right)\right]\right)^n \exp(-\lambda a)$$

$$P\left(-\frac{n}{\lambda}\log\left(1 - \lambda \mathbb{E}\left[h\right]\right) - \sum_{i \in [n]}\left(h_i + \sum_{\ell=2}^{2L}\frac{1}{\ell}(\lambda h_i)^\ell\right) > a\right) \leq \exp(-\lambda a) \quad \text{(Use of Equation (43))}$$

Solving for $\delta = \exp(-\lambda a)$, we get:

$$P\left(-\frac{n}{\lambda}\log\left(1 - \lambda \mathbb{E}\left[h\right]\right) - \sum_{i \in [n]}\left(h_i + \sum_{\ell=2}^{2L}\frac{1}{\ell}(\lambda h_i)^\ell\right) > \frac{\ln(1/\delta)}{\lambda}\right) \leq \delta$$

$$P\left(-\frac{1}{\lambda}\log\left(1 - \lambda \mathbb{E}\left[h\right]\right) - \frac{1}{n}\sum_{i \in [n]}\left(h_i + \sum_{\ell=2}^{2L}\frac{\lambda^\ell}{\ell}h_i^\ell\right) > \frac{\ln(1/\delta)}{\lambda n}\right) \leq \delta$$

$$P\left(-\frac{1}{\lambda}\log\left(1 - \lambda \mathbb{E}\left[h\right]\right) - \hat{R}_n^h(\pi) - \sum_{\ell=2}^{2L}\frac{\lambda^{\ell-1}}{\ell}\hat{\mathcal{M}}_n^{h,\ell}(\pi) > \frac{\ln(1/\delta)}{\lambda n}\right) \leq \delta$$

$$P\left(-\frac{1}{\lambda}\log\left(1 - \lambda \mathbb{E}\left[h\right]\right) > \hat{R}_n^h(\pi) + \sum_{\ell=2}^{2L}\frac{\lambda^{\ell-1}}{\ell}\hat{\mathcal{M}}_n^{h,\ell}(\pi)\frac{\ln(1/\delta)}{\lambda n}\right) \leq \delta.$$

This means that the following, complementary event will hold with probability at least $1 - \delta$:

$$-\frac{1}{\lambda}\log\left(1 - \lambda \mathbb{E}\left[h\right]\right) \leq \hat{R}_n^h(\pi) + \sum_{\ell=2}^{2L}\frac{\lambda^{\ell-1}}{\ell}\hat{\mathcal{M}}_n^{h,\ell}(\pi)\frac{\ln(1/\delta)}{\lambda n}.$$

$\psi_\lambda$ being a non-decreasing function, applying it to the two sides of this inequality gives us:

$$\mathbb{E}\left[h\right] \leq \psi_\lambda\left(\hat{R}_n^h(\pi) + \sum_{\ell=2}^{2L}\frac{\lambda^{\ell-1}}{\ell}\hat{\mathcal{M}}_n^{h,\ell}(\pi) + \frac{\ln(1/\delta)}{\lambda n}\right).$$

Finally, $h$ satisfies **(C1)**, this means that the bound is also an upper bound on the true risk, giving:

$$R(\pi) \leq \psi_\lambda\left(\hat{R}_n^h(\pi) + \sum_{\ell=2}^{2L}\frac{\lambda^{\ell-1}}{\ell}\hat{\mathcal{M}}_n^{h,\ell}(\pi) + \frac{\ln(1/\delta)}{\lambda n}\right),$$

which concludes the proof. $\qquad\square$

## F.2 Proof of the impact of $L$ on the bound's tightness (Proposition 2)

> **Proposition** (Impact of $L$ on the bound's tightness). *Let $\pi \in \Pi$, $\delta \in (0,1]$, $\lambda > 0$, $L \geq 1$ and $h$ satisfying (C1). Let*
>
> $$U_L^{\lambda,h}(\pi) = \psi_\lambda \left( \hat{R}_n^h(\pi) + \frac{\ln(1/\delta)}{\lambda n} + \sum_{\ell=2}^{2L} \frac{\lambda^{\ell-1}}{\ell} \hat{\mathcal{M}}_n^{h,\ell}(\pi) \right)$$
>
> *be the upper bound in Equation (6). Then,*
>
> $$\lambda \leq \min_{i \in [n]} \left\{ \frac{2L+2}{(2L+1)|h_i|} \right\} \implies U_{L+1}^{\lambda,h}(\pi) \leq U_L^{\lambda,h}(\pi). \tag{44}$$
>
> *which implies that:*
>
> $$\lambda \leq \min_{i \in [n]} \left\{ \frac{1}{|h_i|} \right\} \implies U_L^{\lambda,h}(\pi) \text{ is a decreasing function w.r.t } L.$$

*Proof.* We want to prove the implication (44) from which the condition on the decreasing nature of our bound will follow. Indeed, Let us suppose that (44) is true, we have:

$$\lambda \leq \min_{i \in [n]} \left\{ \frac{1}{|h_i|} \right\} \implies \forall L \geq 1, \quad \lambda \leq \min_{i \in [n]} \left\{ \frac{2L+2}{(2L+1)|h_i|} \right\}$$

$$\implies \forall L \geq 1, \quad U_{L+1}^{\lambda,h}(\pi) \leq U_L^{\lambda,h}(\pi) \quad \text{(Using (44))}$$

$$\implies U_L^{\lambda,h}(\pi) \text{ is a decreasing function w.r.t } L.$$

Now let us prove the implication in (44). We have for any $L \geq 1$:

$$U_{L+1}^{\lambda,h}(\pi) \leq U_L^{\lambda,h}(\pi) \iff \sum_{\ell=2L+1}^{2L+2} \frac{\lambda^{\ell-1}}{\ell} \hat{\mathcal{M}}_n^{h,\ell}(\pi) \leq 0$$

$$\iff \frac{\lambda^{2L}}{n} \sum_{i=1}^n h_i^{2L+1} \left( \frac{1}{2L+1} + \frac{\lambda h_i}{2L+2} \right) \leq 0$$

As $h_i \leq 0$, we can ensure this inequality by choosing a $\lambda$ that verifies:

$$\forall i \in [n], \quad \lambda \leq \left\{ \frac{2L+2}{(2L+1)|h_i|} \right\} \iff \lambda \leq \min_{i \in [n]} \left\{ \frac{2L+2}{(2L+1)|h_i|} \right\}$$

which concludes the proof. $\qquad\square$

## F.3 Comparisons of the bounds $U_L^\lambda$ (Proposition 3)

We compare the bounds evaluated in their optimal regularisation function $h$. We start by stating the proposition and proving it.

> **Proposition.** *Let $\pi \in \Pi$, and $\lambda > 0$, we define:*
>
> $$U_L^\lambda(\pi) = \min_h U_L^{\lambda,h}(\pi).$$
>
> *Then, for any $\lambda > 0$, it holds that for any $L > 1$:*
>
> $$U_L^\lambda(\pi) \leq U_1^\lambda(\pi).$$
>
> *In particular, $\forall \lambda > 0$:*
>
> $$U_\infty^\lambda(\pi) \leq U_1^\lambda(\pi), \tag{45}$$

*Proof.* Let $\pi \in \Pi$, $\lambda > 0$ and

$$U_L^\lambda(\pi) = \min_h U_L^{\lambda,h}(\pi).$$

We can prove (see Appendix F.4) that:

$$U_1^\lambda(\pi) = U_1^{\lambda,h_{*,1}}(\pi) = \psi_\lambda\left(\hat{R}_n^{h_{*,1}}(\pi) + \frac{\lambda}{2}\hat{\mathcal{M}}_n^{h_{*,1},2}(\pi) + \frac{\ln(1/\delta)}{\lambda n}\right)$$

with:

$$h_{*,1}(p,q,c) = -\min(|c|p/q, 1/\lambda),$$

and that (see Appendix F.7):

$$U_\infty^\lambda(\pi) = \psi_\lambda\left(\hat{R}_n^\lambda(\pi) + \frac{\ln(1/\delta)}{\lambda n}\right).$$

From Proposition 2, we have that for any $h$:

$$\lambda \leq \min_{i\in[n]}\left\{\frac{1}{|h_i|}\right\} \implies U_L^{\lambda,h}(\pi) \text{ is a decreasing function w.r.t } L.$$

It appears that the optimal function $h_{*,1}$ respects this condition, as by definition:

$$\min_{i\in[n]}\left\{\frac{1}{|(h_{*,1})_i|}\right\} \geq \lambda,$$

meaning that:

$$U_L^{\lambda,h_{*,1}}(\pi) \text{ is a decreasing function w.r.t } L.$$

This result suggests that the Empirical Second Moment bound, evaluated in its optimal function $h_{*,1}$, is always bigger than bounds with additional moments (evaluated in the same $h_{*,1}$). This leads us to the result wanted, as for any $L > 1$:

$$U_L^\lambda(\pi) = \min_h U_L^{\lambda,h}(\pi) \leq U_L^{\lambda,h_{*,1}}(\pi) \leq U_1^{\lambda,h_{*,1}}(\pi) = U_1^\lambda(\pi).$$

In particular, we get:

$$U_\infty^\lambda(\pi) \leq U_1^\lambda(\pi),$$

which ends the proof. $\qquad\square$

This means that $U_\infty^\lambda$ is tighter than $U_1^\lambda$, and thus can also be tighter than empirical Bernstein.

### F.4   Proof of the optimality of global clipping for Corollary 4

> **Proposition** (Optimal $h$ for $L = 1$). *Let $\lambda > 0$. The function $h$ that minimizes the bound for $L = 1$, giving the tightest result is:*
>
> $$\forall i, \quad h_i = h(\pi(a_i|x_i), \pi_0(a_i|x_i), c_i)) = -\min\left\{\frac{\pi(a_i|x_i)}{\pi_0(a_i|x_i)}|c_i|, \frac{1}{\lambda}\right\}$$
>
> *This means that when the costs are binary, we obtain the classical Clipping estimator of parameter $1/\lambda$:*
>
> $$h_i = \min\left\{\frac{\pi(a_i|x_i)}{\pi_0(a_i|x_i)}, \frac{1}{\lambda}\right\}c_i.$$

*Proof.* We want to look for the value of $h$ that minimizes the bound. Formally, by fixing all variables of the bound, this problem reduces to:

$$\operatorname*{argmin}_{h\in\textbf{(C1)}} \hat{R}_n^h(\pi) + \frac{\lambda}{2}\hat{\mathcal{M}}_n^{h,2}(\pi) = \operatorname*{argmin}_{h\in\textbf{(C1)}} \frac{1}{n}\sum_{i=1}^{n}\left(h_i + \frac{\lambda}{2}h_i^2\right).$$

The objective decomposes across data points, so we can solve it for every $h_i$ independently. Let us fix a $j \in [n]$, the following problem:

$$\underset{h_j \in \mathbb{R}}{\operatorname{argmin}} \, \hat{R}_n^h(\pi) + \frac{\lambda}{2}\hat{\mathcal{M}}_n^{h,2}(\pi) = \underset{h_j \in \mathbb{R}}{\operatorname{argmin}} \left\{ h_j + \frac{\lambda}{2}h_j^2 \right\}$$

$$\text{subject to} \quad h_j \geq \frac{\pi(a_j|x_j)}{\pi_0(a_j|x_j)}c_j$$

is strongly convex in $h_j$. We write the KKT conditions for $h_j$ to be optimal; there exists $\alpha^*$ that verifies:

$$1 + \lambda h_j - \alpha^* = 0 \tag{46}$$

$$\alpha^* \geq 0 \tag{47}$$

$$\alpha^* \left( \frac{\pi(a_j|x_j)}{\pi_0(a_j|x_j)}c_j - h_j \right) = 0 \tag{48}$$

$$h_j \geq \frac{\pi(a_j|x_j)}{\pi_0(a_j|x_j)}c_j \tag{49}$$

We study the two following two cases:

**Case 1:** $h_j \leq -\frac{1}{\lambda}$ :
we have $\alpha^* = 1 + \lambda h_j \leq 0 \implies \alpha^* = 0$, meaning that:

$$h_j = -\frac{1}{\lambda}$$

**Case 2:** $h_j > -\frac{1}{\lambda}$ :
we have $\alpha^* = 1 + \lambda h_j > 0$, which combined to condition (36) gives:

$$h_j = \frac{\pi(a_j|x_j)}{\pi_0(a_j|x_j)}c_j.$$

The two results combined mean that we always have:

$$h_j \geq -\frac{1}{\lambda}, \text{ and whenever } h_j > -\frac{1}{\lambda} \implies h_j = \frac{\pi(a_j|x_j)}{\pi_0(a_j|x_j)}c_j.$$

We deduce that $h_j$ has the following form:

$$h_j = h(\pi(a_j|x_j), \pi_0(a_j|x_j), c_j) = -\min \left\{ \frac{\pi(a_j|x_j)}{\pi_0(a_j|x_j)}|c_j|, \frac{1}{\lambda} \right\} \tag{50}$$

$$\alpha^* = 1 - \lambda \min \left\{ \frac{\pi(a_j|x_j)}{\pi_0(a_j|x_j)}|c_j|, \frac{1}{\lambda} \right\} \tag{51}$$

These values verify the KKT conditions. As the problem is strongly convex, $h_j$ has a unique possible value and must be equal to equation (38). The form of $h_j$ is a global clipping that includes the cost in the function as well. In the case where the cost function $c$ is binary:

$$\forall i \quad c_i \in \{-1, 0\},$$

we recover the classical Clipping with parameter $1/\lambda$ as an optimal solution for $h$:

$$h_j = \min \left\{ \frac{\pi(a_j|x_j)}{\pi_0(a_j|x_j)}, \frac{1}{\lambda} \right\} c_j.$$

$\square$

### F.5 Comparison with empirical Bernstein

We begin by comparing the Second Moment Bound with Swaminathan and Joachims [55]'s bound as they both manipulate similar quantities. The bound of [55] uses the Empirical Bernstein bound of [36] applied to the Clipping Estimator. We recall its expression below for a parameter $M > 0$:

$$\hat{R}_n^M(\pi) = \frac{1}{n} \sum_{i=1}^n \min \left\{ \frac{\pi(a_i|x_i)}{\pi_0(a_i|x_i)}, M \right\} c_i.$$

We also give below the Empirical Bernstein Bound applied to this estimator:

---

**Proposition** (Empirical Bernstein for Clipping of [55]). *Let $\pi \in \Pi$, $\delta \in (0,1]$ and $M > 0$. Then it holds with probability at least $1 - \delta$ that*

$$R(\pi) \le \hat{R}_n^M(\pi) + \sqrt{\frac{2\hat{V}_n^M(\pi)\ln(2/\delta)}{n}} + \frac{7M\ln(2/\delta)}{3(n-1)}, \tag{52}$$

*with $\hat{V}_n^M(\pi)$ the empirical variance of the clipping estimator.*

---

We are usually interested in the case where $\pi$ and $\pi_0$ are different, leading to substantial importance weights. In this practical scenario, the variance and the second moment are of the same magnitude of $M$. Indeed, one can see it from the following equality:

$$\underbrace{\hat{V}_n^M(\pi)}_{\mathcal{O}(M)} = \underbrace{\hat{\mathcal{M}}_n^{M,2}(\pi)}_{\mathcal{O}(M)} - \underbrace{\left(\hat{R}_n^M(\pi)\right)^2}_{\mathcal{O}(\bar{c}^2)}$$

$$\approx \underbrace{\hat{\mathcal{M}}_n^{M,2}(\pi)}_{\mathcal{O}(M)} \quad (M \gg \bar{c}^2 = o(1).)$$

This means that in practical scenarios, the empirical variance and the empirical second moment are approximately the same. Recall that the Second Moment Bound works for any regularizer $h$, As Clipping satisfies **(C1)**, we give the Second Moment Upper of Corollary 4 with Clipping below:

$$\psi_\lambda\left(\hat{R}_n^M(\pi) + \frac{\lambda}{2}\hat{\mathcal{M}}_n^{M,2}(\pi) + \frac{\ln(1/\delta)}{\lambda n}\right) \le \hat{R}_n^M(\pi) + \frac{\lambda}{2}\hat{\mathcal{M}}_n^{M,2}(\pi) + \frac{\ln(1/\delta)}{\lambda n} \quad (\psi_\lambda(x) \le x, \forall x)$$

$$\le \hat{R}_n^M(\pi) + \frac{\lambda}{2}\hat{\mathcal{M}}_n^{M,2}(\pi) + \frac{\ln(1/\delta)}{\lambda n}.$$

Choosing a $\lambda \approx \sqrt{2\ln(1/\delta)/(n\hat{\mathcal{M}}_n^{M,2}(\pi))}$ gives us an upper bound that is close to:

$$\hat{R}_n^M(\pi) + \frac{\lambda}{2}\hat{\mathcal{M}}_n^{M,2}(\pi) + \frac{\ln(1/\delta)}{\lambda n} \approx \hat{R}_n^M(\pi) + \sqrt{\frac{2\hat{\mathcal{M}}_n^{M,2}(\pi)\ln(1/\delta)}{n}}$$

$$\approx \hat{R}_n^M(\pi) + \sqrt{\frac{2\hat{V}_n^M(\pi)\ln(1/\delta)}{n}}$$

$$\le \hat{R}_n^M(\pi) + \sqrt{\frac{2\hat{V}_n^M(\pi)\ln(2/\delta)}{n}} + \frac{7M\ln(2/\delta)}{3(n-1)}.$$

This means that in practical scenarios, and with a good choice of $\lambda \sim \mathcal{O}(1/\sqrt{n})$, the Second Moment bound would be better than the Empirical Bernstein bound, and this difference will be even greater when $M \gg 1$. This is aligned with our experiments, where we see that the new Second Moment bound is much tighter in practice. This also confirms that the Logarithmic smoothing bound is even tighter, because it is smaller than the Second Moment bound as stated in Proposition 3.

### F.6 Proof of the $L \to \infty$ bound (Corollary 5)

> **Proposition** (Empirical Logarithmic Smoothing bound with $L \to \infty$). *Let $\pi \in \Pi$, $\delta \in (0, 1]$ and $\lambda > 0$. Then it holds with probability at least $1 - \delta$ that*
> $$R(\pi) \leq \psi_\lambda\left( -\frac{1}{n} \sum_{i=1}^n \frac{1}{\lambda} \log\left(1 - \lambda h_i\right) + \frac{\ln(1/\delta)}{\lambda n} \right).$$

Taking the limit of $L$ naively recovers this form of the bound, but imposes a condition on $\lambda$ for the bound to converge. We instead, take another path of proof that does not impose any condition on $\lambda$, developed below. The main idea is to take the limit of $L$ to recover the variable to use along Chernoff.

*Proof.* Recall that for the proof of the Empirical moments bounds, we used the following random variable defined with $\lambda > 0$:
$$X_i = -\frac{1}{\lambda} \log\left(1 - \lambda \mathbb{E}\left[h\right]\right) - h_i - \sum_{\ell=2}^{2L} \frac{1}{\ell} (\lambda h_i)^\ell,$$

combined with Chernoff Inequality (Lemma 12) to prove our bound. If we take the limit $L \to \infty$ for our random variable, we obtain the following random variable:
$$\tilde{X}_i = -\frac{1}{\lambda} \log\left(1 - \lambda \mathbb{E}\left[h\right]\right) + \frac{1}{\lambda} \log\left(1 - \lambda h_i\right)$$
$$= \frac{1}{\lambda} \log\left( \frac{1 - \lambda h_i}{1 - \lambda \mathbb{E}\left[h\right]} \right).$$

We use the random variable $\tilde{X}_i$ with the Chernoff Inequality. For any $a \in \mathbb{R}$, we have:
$$P\left( \sum_{i \in [n]} \tilde{X}_i > a \right) \leq \left( \mathbb{E}\left[ \exp\left( \lambda \tilde{X}_1 \right) \right] \right)^n \exp(-\lambda a)$$
$$P\left( -\frac{n}{\lambda} \log\left(1 - \lambda \mathbb{E}\left[h\right]\right) + \sum_{i \in [n]} \left( \frac{1}{\lambda} \log\left(1 - \lambda h_i\right) \right) > a \right) \leq \left( \mathbb{E}\left[ \exp\left( \lambda \tilde{X}_1 \right) \right] \right)^n \exp(-\lambda a)$$

On the other hand, we have:
$$\mathbb{E}\left[ \exp\left( \lambda \tilde{X}_1 \right) \right] = \frac{\mathbb{E}\left[1 - \lambda h_i\right]}{1 - \lambda \mathbb{E}\left[h\right]} = 1.$$

Using this equality and solving for $\delta = \exp(-\lambda a)$, we get:
$$P\left( -\frac{n}{\lambda} \log\left(1 - \lambda \mathbb{E}\left[h\right]\right) + \sum_{i \in [n]} \left( \frac{1}{\lambda} \log\left(1 - \lambda h_i\right) \right) > \frac{\ln(1/\delta)}{\lambda} \right) \leq \delta$$
$$P\left( -\frac{1}{\lambda} \log\left(1 - \lambda \mathbb{E}\left[h\right]\right) + \frac{1}{n} \sum_{i \in [n]} \frac{1}{\lambda} \log\left(1 - \lambda h_i\right) > \frac{\ln(1/\delta)}{\lambda n} \right) \leq \delta$$

This means that the following, complementary event will hold with probability at least $1 - \delta$:
$$-\frac{1}{\lambda} \log\left(1 - \lambda \mathbb{E}\left[h\right]\right) \leq -\frac{1}{n} \sum_{i=1}^n \frac{1}{\lambda} \log\left(1 - \lambda h_i\right) + \frac{\ln(1/\delta)}{\lambda n}.$$

$\psi_\lambda$ being a non-decreasing function, applying it to the two sides of this inequality gives us:
$$\mathbb{E}\left[h\right] \leq \psi_\lambda\left( -\frac{1}{n} \sum_{i=1}^n \frac{1}{\lambda} \log\left(1 - \lambda h_i\right) + \frac{\ln(1/\delta)}{\lambda n} \right).$$

As $h$ satisfies (**C1**), we obtain the required inequality:
$$R(\pi) \leq \psi_\lambda\left( -\frac{1}{n} \sum_{i=1}^n \frac{1}{\lambda} \log\left(1 - \lambda h_i\right) + \frac{\ln(1/\delta)}{\lambda n} \right).$$

and conclude the proof. $\qquad\square$

### F.7 Proof of the optimality of IPS for Corollary 5

> **Proposition** (Optimal $h$ for $L \to \infty$)**.** *Let $\lambda > 0$. The function $h$ that minimizes the bound for $L \to \infty$, giving the tightest result is:*
>
> $$\forall i, \quad h_i = h(\pi(a_i|x_i), \pi_0(a_i|x_i), c_i)) = \frac{\pi(a_i|x_i)}{\pi_0(a_i|x_i)} c_i$$

*Proof.* The proof of this proposition is quite simple. The function:

$$f(x) = -\log(1 - \lambda x)$$

is increasing. This means that the lowest possible value of $h_i$ ensures the tightest result. As our variables $h_i$ verifies **(C1)**, we recover IPS as an optimal choice for this bound. $\qquad\square$

### F.8 Comparison with the IX bound (Proposition 8)

We now attack the recently derived IX bound in Gabbianelli et al. [21] and show that our newly proposed bound dominates it in all scenarios.

> **Proposition** (Comparison with IX [21])**.** *Let $\pi \in \Pi$, $\delta \in\,]0, 1]$ and $\lambda > 0$, the IX bound from [21] states that we have with at least probability $1 - \delta$:*
>
> $$R(\pi) \leq \hat{R}_n^{\lambda\text{-IX}}(\pi) + \frac{\ln(1/\delta)}{\lambda n} \tag{53}$$
>
> *with:*
>
> $$\hat{R}_n^{\lambda\text{-IX}}(\pi) = \frac{1}{n} \sum_{i=1}^{n} \frac{\pi(a_i|x_i)}{\pi_0(a_i|x_i) + \lambda/2} c_i.$$
>
> *Let $U_{\text{IX}}^\lambda(\pi)$ be the IX upper bound defined above, we have for any $\lambda > 0$:*
>
> $$U_\infty^\lambda(\pi) \leq U_{\text{IX}}^\lambda(\pi). \tag{54}$$

*Proof.* Let $\pi \in \Pi$, $\delta \in\,]0, 1]$ and $\lambda > 0$. Recall that $U_\infty^\lambda(\pi) = \psi_\lambda\left(\hat{R}_n^\lambda(\pi) + \frac{\ln(1/\delta)}{\lambda n}\right)$. We have:

$$\psi_\lambda\left(\hat{R}_n^\lambda(\pi) + \frac{\ln(1/\delta)}{\lambda n}\right) \leq \hat{R}_n^\lambda(\pi) + \frac{\ln(1/\delta)}{\lambda n} \quad (\forall x, \psi_\lambda(x) \leq x)$$

$$\leq -\frac{1}{n} \sum_{i=1}^{n} \frac{1}{\lambda} \log(1 - \lambda w_\pi(x_i, a_i) c_i) + \frac{\ln(1/\delta)}{\lambda n}.$$

Using the inequality $\log(1 + x) \geq \frac{x}{1+x/2}$ for all $x > 0$, we get:

$$U_\infty^\lambda(\pi) \leq -\frac{1}{n} \sum_{i=1}^{n} \frac{1}{\lambda} \log(1 - \lambda w_\pi(x_i, a_i) c_i) + \frac{\ln(1/\delta)}{\lambda n}$$

$$\leq \frac{1}{n} \sum_{i=1}^{n} \frac{w_\pi(x_i, a_i)}{1 - \lambda w_\pi(x_i, a_i) c_i/2} c_i + \frac{\ln(1/\delta)}{\lambda n} \quad \left(\log(1+x) \geq \frac{x}{1+x/2}\right)$$

$$\leq \frac{1}{n} \sum_{i=1}^{n} \frac{\pi(a_i|x_i)}{\pi_0(a_i|x_i) - \lambda\pi(a_i|x_i)c_i/2} c_i + \frac{\ln(1/\delta)}{\lambda n}$$

$$\leq \frac{1}{n} \sum_{i=1}^{n} \frac{\pi(a_i|x_i)}{\pi_0(a_i|x_i) + \lambda/2} c_i + \frac{\ln(1/\delta)}{\lambda n} \quad (-\pi(a_i|x_i)c_i \leq 1 \text{ and } c_i \leq 0)$$

$$\leq \hat{R}_n^{IX-\lambda}(\pi) + \frac{\ln(1/\delta)}{\lambda n} = U_{IX}^\lambda(\pi),$$

which ends the proof. $\qquad\square$

The result states the dominance of the LS bound compared to IX. The proof of this result also gives us insight on when the LS bound will be much tighter than IX. Indeed, to obtain the IX bound, LS bound is loosened through 3 steps:

1. The use of $\psi_\lambda(x) \le x, \forall x$.
2. The use of $\log(1 + \lambda x) \ge \frac{\lambda x}{1 + \lambda x/2}, \forall x \ge 0$.
3. The use of $-\pi(a_i|x_i)c_i \le 1, \forall i \in [n]$.

The two first inequalities are loose when $\lambda \sim 1/\sqrt{n}$ is not too small, which means that LS will be much better in problems with few samples. The third inequality is loose when $\pi$ is not a peaked policy or the cost is way less than 1. Even if LS bound is always smaller than IX, LS will give way better result if the number of samples is small, and/or the policy evaluated is diffused.

# G  Proofs of OPS and OPL

## G.1  OPS: Proof of suboptimality bound (Proposition 9)

> **Proposition** (Suboptimality of our selection strategy in (20)). *Let $\lambda > 0$ and $\delta \in (0, 1]$. Then, it holds with probability at least $1 - \delta$ that*
>
> $$0 \le R(\hat{\pi}_n^{\text{s}}) - R(\pi_*^{\text{s}}) \le \lambda \mathcal{S}_\lambda(\pi_*^{\text{s}}) + \frac{2\ln(2|\Pi_{\text{s}}|/\delta)}{\lambda n} \, ,$$
>
> *where $\pi_*^{\text{s}}$ and $\hat{\pi}_n^{\text{s}}$ are defined in (19) and (20), and*
>
> $$\mathcal{S}_\lambda(\pi) = \mathbb{E}\left[ (w_\pi(x,a)c)^2 / (1 - \lambda w_\pi(x,a)c) \right].$$
>
> *In addition, our upper bound is always finite as:*
>
> $$\lambda \mathcal{S}_\lambda(\pi) = \lambda \mathbb{E}\left[ \frac{(w_\pi(x,a)c)^2}{1 - \lambda w_\pi(x,a)c} \right] \le \min\left\{ |R(\pi)|, \lambda \mathbb{E}\left[ (w_\pi(x,a)c)^2 \right] \right\} \le |R(\pi)|.$$

*Proof.* To prove this bound on the suboptimality of our selection method, we need both an upper bound and a lower bound on the true risk using the LS estimator. Luckily, we already have derived them in **??** . For a fixed $\lambda$, taking a union of the two bounds over the cardinal of the finite policy class $|\Pi_s|$, we get the following holding with probability at least $1 - \delta$ for all $\pi \in \Pi_s$:

$$R(\pi) - \hat{R}_n^\lambda(\pi) \le \frac{\ln(2|\Pi_s|/\delta)}{\lambda n} \, , \qquad \text{and} \qquad \hat{R}_n^\lambda(\pi) - R(\pi) \le \lambda \mathcal{S}_\lambda(\pi) + \frac{\ln(2|\Pi_s|/\delta)}{\lambda n} \, .$$

As $\hat{\pi}_n^{\text{s}} \in \Pi_s$ and by definition of $\hat{\pi}_n^{\text{s}}$ (minimizer of $\hat{R}_n^\lambda(\pi)$), we have:

$$R(\hat{\pi}_n^{\text{s}}) \le \hat{R}_n^\lambda(\hat{\pi}_n^{\text{s}}) + \frac{\ln(2|\Pi_s|/\delta)}{\lambda n} \le \hat{R}_n^\lambda(\hat{\pi}_*^{\text{s}}) + \frac{\ln(2|\Pi_s|/\delta)}{\lambda n}.$$

Using the lower bound on the risk of $R(\hat{\pi}_*^{\text{s}})$, we have:

$$R(\hat{\pi}_n^{\text{s}}) \le \hat{R}_n^\lambda(\hat{\pi}_*^{\text{s}}) + \frac{\ln(2|\Pi_s|/\delta)}{\lambda n}$$

$$\le R(\hat{\pi}_*^{\text{s}}) + \lambda \mathcal{S}_\lambda(\hat{\pi}_*^{\text{s}}) + \frac{2\ln(2|\Pi_s|/\delta)}{\lambda n}.$$

which gives us the suboptimality upper bound:

$$0 \le R(\hat{\pi}_n^{\text{s}}) - R(\pi_*^{\text{s}}) \le \lambda \mathcal{S}_\lambda(\pi_*^{\text{s}}) + \frac{2\ln(2|\Pi_{\text{s}}|/\delta)}{\lambda n} \, .$$

Note that:

$$\lambda \mathcal{S}_\lambda(\pi) = \lambda \mathbb{E}\left[ \frac{(w_\pi(x,a)c)^2}{1 - \lambda w_\pi(x,a)c} \right] \le \min\left\{ |R(\pi)|, \lambda \mathbb{E}\left[ (w_\pi(x,a)c)^2 \right] \right\},$$

always ensuring a finite bound. $\qquad \square$

### G.2 OPL: Proof of PAC-Bayesian LS-LIN bound (Proposition 10)

**Proposition** (PAC-Bayes learning bound for $\hat{R}_n^{\lambda-\texttt{LIN}}$). *Given a prior $P \in \mathcal{P}(\Theta)$, $\delta \in (0, 1]$ and $\lambda > 0$, the following holds with probability at least $1 - \delta$:*

$$\forall Q \in \mathcal{P}(\Theta), \quad R(\pi_Q) \leq \psi_\lambda \left( \hat{R}_n^{\lambda-\texttt{LIN}}(\pi_Q) + \frac{\mathcal{KL}(Q||P) + \ln \frac{1}{\delta}}{\lambda n} \right)$$

*Proof.* To prove this proposition, we can either take the path of High Order Empirical moments as for Pessimistic OPE, or we can prove it directly. We provide here a simple proof of this proposition using ideas from Alquier [1, Corollary 2.5]. Let:

$$d_\theta(a|x) = \mathbb{1}\left[ f_\theta(x) = a \right], \forall (x, a) \in \mathcal{X} \times \mathcal{A}, \tag{55}$$

it means that:

$$\pi_Q(a|x) = \mathbb{E}_{\theta \sim Q}\left[ d_\theta(a|x) \right], \forall (x, a) \in \mathcal{X} \times \mathcal{A}.$$

Recall that to prove a PAC-Bayesian generalization bound, one can rely on the Change of measure Lemma (Lemma 14). For any $\lambda > 0$, the adequate function $g$ to consider is:

$$\begin{aligned}
g(\theta, \mathcal{D}_n) &= \sum_{i=1}^n \left( -\log(1 - \lambda R(d_\theta)) + \log\left( 1 - \lambda \frac{d_\theta(a_i|x_i)c_i}{\pi_0(a_i|x_i)} \right) \right) \\
&= \sum_{i=1}^n \log\left( \frac{1 - \lambda \frac{d_\theta(a_i|x_i)c_i}{\pi_0(a_i|x_i)}}{1 - \lambda R(d_\theta)} \right).
\end{aligned}$$

By exploiting the i.i.d. nature of the data and exchanging the order of expectations ($P$ is independent of $\mathcal{D}_n$), we can naturally prove that:

$$\begin{aligned}
\Psi_g &= \mathbb{E}_P \left[ \prod_{i=1}^n \mathbb{E} \left[ \exp\left( \log\left( \frac{1 - \lambda \frac{d_\theta(a_i|x_i)c_i}{\pi_0(a_i|x_i)}}{1 - \lambda R(d_\theta)} \right) \right) \right] \right] \\
&= \mathbb{E}_P \left[ \prod_{i=1}^n \mathbb{E} \left[ \frac{1 - \lambda \frac{d_\theta(a_i|x_i)c_i}{\pi_0(a_i|x_i)}}{1 - \lambda R(d_\theta)} \right] \right] \\
&= \mathbb{E}_P \left[ \prod_{i=1}^n \frac{1 - \lambda R(d_\theta)}{1 - \lambda R(d_\theta)} \right] = 1.
\end{aligned}$$

Injecting $\Psi_g$ in Lemma 14, gives:

$$\begin{aligned}
\mathbb{E}_{\theta \sim Q}\left[ -\log(1 - \lambda R(d_\theta)) \right] &\leq \frac{1}{n} \sum_{i=1}^n \mathbb{E}_{\theta \sim Q}\left[ -\log\left( 1 - \lambda \frac{d_\theta(a_i|x_i)c_i}{\pi_0(a_i|x_i)} \right) \right] + \frac{\mathcal{KL}(Q||P) + \ln \frac{1}{\delta}}{n} \\
&\leq \frac{1}{n} \sum_{i=1}^n \mathbb{E}_{\theta \sim Q}\left[ -d_\theta(a_i|x_i)\log\left( 1 - \lambda \frac{c_i}{\pi_0(a_i|x_i)} \right) \right] + \frac{\mathcal{KL}(Q||P) + \ln \frac{1}{\delta}}{n} \\
&\leq -\frac{1}{n} \sum_{i=1}^n \pi_Q(a_i|x_i)\log\left( 1 - \lambda \frac{c_i}{\pi_0(a_i|x_i)} \right) + \frac{\mathcal{KL}(Q||P) + \ln \frac{1}{\delta}}{n} \\
&\leq \lambda \hat{R}_n^{\lambda-\texttt{LIN}}(\pi_Q) + \frac{\mathcal{KL}(Q||P) + \ln \frac{1}{\delta}}{n}.
\end{aligned}$$

From the convexity of $x \to -\log(1 + x)$, we have:

$$-\frac{1}{\lambda}\log\left( 1 - \lambda R(\pi_Q) \right) \leq \frac{1}{\lambda}\mathbb{E}_{\theta \sim Q}\left[ -\log(1 - \lambda R(d_\theta)) \right] \leq \hat{R}_n^{\lambda-\texttt{LIN}}(\pi_Q) + \frac{\mathcal{KL}(Q||P) + \ln \frac{1}{\delta}}{\lambda n}.$$

Applying the increasing function $\psi_\lambda$ of Equation (5) to both sides concludes the proof. $\square$

### G.3 OPL: Proof of PAC-Bayesian suboptimality bound (Proposition 11)

> **Proposition** (Suboptimality of the learning strategy in (27)). *Let $\lambda > 0$, $P \in \mathcal{L}(\Theta)$ and $\delta \in (0,1]$. Then, it holds with probability at least $1 - \delta$ that*
>
> $$0 \leq R(\hat{\pi}_{Q_n}) - R(\pi_{Q^*}) \leq \lambda \mathcal{S}_{\lambda}^{\texttt{LIN}}(\pi_{Q^*}) + \frac{2\left(\mathcal{KL}(Q^*||P) + \ln(2/\delta)\right)}{\lambda n},$$
>
> *where*
>
> $$\mathcal{S}_{\lambda}^{\texttt{LIN}}(\pi) = \mathbb{E}\left[\frac{\pi(a|x)c^2}{\pi_0^2(a|x) - \lambda \pi_0(a|x)c}\right].$$
>
> *In addition, our upper bound is always finite as:*
>
> $$\lambda \mathcal{S}_{\lambda}^{\texttt{LIN}}(\pi) \leq \min\left\{|R(\pi)|, \lambda \mathbb{E}\left[\frac{\pi(a|x)c^2}{\pi_0^2(a|x)}\right]\right\} \leq |R(\pi)|.$$

*Proof.* To prove this bound on the suboptimality of our learning strategy, we need both a PAC-Bayesian upper bound and a lower bound on the true risk using the LS-LIN estimator. Luckily, we already have derived an upper bound in Proposition 10, that we linearize here as $\psi_\lambda(x) \leq x$:

$$\forall Q \in \mathcal{P}(\Theta), \quad R(\pi_Q) \leq \hat{R}_n^{\lambda-\texttt{LIN}}(\pi_Q) + \frac{\mathcal{KL}(Q||P) + \ln\frac{1}{\delta}}{\lambda n}.$$

For the lower bound, we rely a second time on the Change of measure Lemma (Lemma 14). For any $\lambda > 0$, we choose the following function $g$:

$$g(\theta, \mathcal{D}_n) = \sum_{i=1}^n \left(-\frac{1}{\lambda}\log\left(1 - \lambda\frac{d_\theta(a_i|x_i)c_i}{\pi_0(a_i|x_i)}\right) - R(d_\theta) - \lambda \mathcal{S}_{\lambda}^{\texttt{LIN}}(d_\theta)\right).$$

By exploiting the i.i.d. nature of the data and exchanging the order of expectations ($P$ is independent of $\mathcal{D}_n$), we can prove that:

$$\Psi_g = \mathbb{E}_P\left[\prod_{i=1}^n \left(\exp\left(-\lambda(R(d_\theta) + \lambda\mathcal{S}_{\lambda}^{\texttt{LIN}}(d_\theta))\right)\mathbb{E}\left[\frac{1}{1 - \lambda\frac{d_\theta(a|x)c}{\pi_0(a|x)}}\right]\right)\right]$$

$$\leq \mathbb{E}_P\left[\prod_{i=1}^n \left(\exp\left(-\lambda(R(d_\theta) + \lambda\mathcal{S}_{\lambda}^{\texttt{LIN}}(d_\theta)) + \mathbb{E}\left[\frac{1}{1 - \lambda\frac{d_\theta(a|x)c}{\pi_0(a|x)}}\right] - 1\right)\right)\right]$$

$$\leq \mathbb{E}_P\left[\prod_{i=1}^n \left(\exp\left(-\lambda(R(d_\theta) + \lambda\mathcal{S}_{\lambda}^{\texttt{LIN}}(d_\theta)) + \mathbb{E}\left[\frac{\lambda d_\theta(a|x)c}{\pi_0(a|x) - \lambda d_\theta(a|x)c}\right]\right)\right)\right]$$

$$\leq \mathbb{E}_P\left[\prod_{i=1}^n \left(\exp\left(-\lambda(R(d_\theta) + \lambda\mathcal{S}_{\lambda}^{\texttt{LIN}}(d_\theta)) + \mathbb{E}\left[\frac{\lambda d_\theta(a|x)c}{\pi_0(a|x) - \lambda c}\right]\right)\right)\right] \quad (d_\theta \text{ is binary.})$$

$$\leq \mathbb{E}_P\left[\prod_{i=1}^n \left(\exp\left(-\lambda^2\mathcal{S}_{\lambda}^{\texttt{LIN}}(d_\theta) + \mathbb{E}\left[\frac{\lambda d_\theta(a|x)c}{\pi_0(a|x) - \lambda c} - \lambda\frac{d_\theta(a|x)c}{\pi_0(a|x)}\right]\right)\right)\right]$$

$$\leq \mathbb{E}_P\left[\prod_{i=1}^n \left(\exp\left(-\lambda^2\mathcal{S}_{\lambda}^{\texttt{LIN}}(d_\theta) + \lambda^2\mathcal{S}_{\lambda}^{\texttt{LIN}}(d_\theta)\right)\right)\right] \leq 1,$$

giving by rearranging terms, the following PAC-Bayesian bound:

$$\forall Q \in \mathcal{P}(\Theta), \quad \hat{R}_n^{\lambda-\texttt{LIN}}(\pi_Q) \leq R(\pi_Q) + \lambda\mathcal{S}_{\lambda}^{\texttt{LIN}}(\pi_Q) + \frac{\mathcal{KL}(Q||P) + \ln(2/\delta)}{\lambda n}.$$

Now we take a union of the the two bounds, for them to hold with probability at least $1 - \delta$ for all $Q$. By definition of $\hat{\pi}_{Q_n}$ (minimizer of the upper bound), we have:

$$R(\hat{\pi}_{Q_n}) \leq \hat{R}_n^{\lambda-\texttt{LIN}}(\hat{\pi}_{Q_n}) + \frac{\mathcal{KL}(Q_n||P) + \ln(2/\delta)}{\lambda n} \leq \hat{R}_n^{\lambda-\texttt{LIN}}(\pi_{Q^*}) + \frac{\mathcal{KL}(Q^*||P) + \ln(2/\delta)}{\lambda n}.$$

Using the lower bound on the risk of $R(\pi_{Q^*})$, we have:

$$R(\hat{\pi}_{Q_n}) \leq \hat{R}_n^{\lambda-\text{LIN}}(\pi_{Q^*}) + \frac{\mathcal{KL}(Q^*||P) + \ln(2/\delta)}{\lambda n}$$
$$\leq R(\pi_{Q^*}) + \lambda \mathcal{S}_\lambda^{\text{LIN}}(\pi_{Q^*}) + \frac{\mathcal{KL}(Q^*||P) + \ln(2/\delta)}{\lambda n}.$$

which gives us the PAC-Bayesian suboptimality upper bound:

$$0 \leq R(\hat{\pi}_{Q_n}) - R(\pi_{Q^*}) \leq \lambda \mathcal{S}_\lambda^{\text{LIN}}(\pi_{Q^*}) + \frac{2\left(\mathcal{KL}(Q^*||P) + \ln(2/\delta)\right)}{\lambda n}.$$

Concluding the proof. □

## H   Experimental design and detailed experiments

All our experiments were conducted on a machine with 16 CPUs. The PAC-Bayesian learning experiments require a moderate amount of computation due to the handling of medium-sized datasets. However, our experiments remain reproducible with minimal computational resources.

### H.1   Off-policy evaluation and selection

#### H.1.1   Datasets

For both our OPE and OPS experiments, we use 11 UCI datasets with different sizes, action spaces and number of features. The statistics of all these datasets are described in Table 3.

Table 3: OPE and OPS: 11 Datasets used from OpenML [8].

| Datasets | $N$ | $K$ | $p$ |
|---|---|---|---|
| **ecoli** | 336 | 8 | 7 |
| **arrhythmia** | 452 | 13 | 279 |
| **micro-mass** | 571 | 20 | 1300 |
| **balance-scale** | 625 | 3 | 4 |
| **eating** | 945 | 7 | 6373 |
| **vehicle** | 846 | 4 | 18 |
| **yeast** | 1484 | 10 | 8 |
| **page-blocks** | 5473 | 5 | 10 |
| **optdigits** | 5620 | 10 | 64 |
| **satimage** | 6430 | 6 | 36 |
| **kropt** | 28 056 | 18 | 6 |

#### H.1.2   (OPE) Tightness of the bounds

**Additional details.**   For these experiments, as we only use oracle policies (faulty policies to log data and we evaluate ideal policies), we use the full 11 datasets without splitting them. The faulty policies are defined exactly as described in the experiments of Kuzborskij et al. [32]. For each datapoint, the behavior (faulty) policy plays an action and we record a cost. The triplets datapoint, action and cost constitute our logged bandit dataset, with which we can compute our estimates and bounds. As we have access to the true label, the original dataset can be used to compute the true risk of any policy.

**Detailed results.**   Evaluating the worst case performance of a policy is done through evaluating risk upper bounds [10, 32]. This means that a better evaluation will solely depend on the tightness of the bounds used. To this end, given a policy $\pi$, we are interested in bounds with a small relative radius $|U(\pi)/R(\pi)-1|$. We compare our newly derived bounds (cIPS-L=1 for $U_1^\lambda$ and LS for $U_\infty^\lambda$ both with $\lambda = 1/\sqrt{n}$) to SNIPS-ES: the Efron Stein bound for Self Normalized IPS [32], cIPS-EB: Empirical Bernstein for Clipping [55] and the recent IX: Implicit Exploration bound [21]. We use all 11 datasets, with different behavior policies ($\tau_0 \in \{0.2, 0.25, 0.3\}$) and different noise levels ($\epsilon \in \{0., 0.1, 0.2\}$) to evaluate ideal policies with different temperatures ($\tau \in \{0.1, 0.2, 0.3, 0.4, 0.5\}$), defining $\sim 500$ different scenarios to validate our findings. In addition to the cumulative distribution of the relative radius of the considered bounds of Figure 2. We give two tables in the following: the average relative

Table 4: OPE: Average relative radius for each datasets

| Datasets | SN-ES | cIPS-EB | IX | cIPS-L=1 | LS |
|---|---|---|---|---|---|
| **ecoli** | 1.00 | 1.00 | 0.676 | 0.752 | **0.573** |
| **arrhythmia** | 1.00 | 1.00 | 0.677 | 0.707 | **0.548** |
| **micro-mass** | 0.962 | 0.840 | 0.394 | 0.346 | **0.311** |
| **balance-scale** | 1.00 | 0.950 | 0.469 | 0.550 | **0.422** |
| **eating** | 0.930 | 0.734 | 0.318 | 0.337 | **0.265** |
| **vehicle** | 0.981 | 0.867 | 0.409 | 0.482 | **0.358** |
| **yeast** | 0.861 | 0.660 | 0.307 | 0.311 | **0.254** |
| **page-blocks** | 0.760 | 0.547 | 0.371 | 0.447 | **0.312** |
| **optdigits** | 0.468 | 0.323 | 0.148 | 0.139 | **0.113** |
| **satimage** | 0.506 | 0.336 | 0.171 | 0.184 | **0.140** |
| **kropt** | 0.224 | 0.161 | 0.087 | 0.066 | **0.060** |

radius of our bounds for each dataset, compiled in Table 4, and the average relative radius of our bounds for each policy evaluated, compiled in Table 5. One can observe that LS always gives the best results no matter the projection. However, the cIPS-L=1 bound is sometimes better than IX, especially when it comes to evaluating diffused policies, see Table 5.

Table 5: OPE: Average relative radiuses for each target policies (ideal policies with different $\tau$)

| $\tau$ | SN-ES | cIPS-EB | IX | cIPS-L=1 | LS |
|---|---|---|---|---|---|
| $\tau = 0.1$ | 0.783 | 0.630 | 0.332 | 0.400 | **0.308** |
| $\tau = 0.2$ | 0.781 | 0.630 | 0.326 | 0.390 | **0.295** |
| $\tau = 0.3$ | 0.782 | 0.668 | 0.353 | 0.389 | **0.297** |
| $\tau = 0.4$ | 0.793 | 0.706 | 0.385 | 0.385 | **0.301** |
| $\tau = 0.5$ | 0.810 | 0.735 | 0.432 | 0.397 | **0.323** |

### H.1.3 (OPS) Find the best, avoid the worst policy

Policy selection aims at identifying the best policy among a set of finite candidates. In practice, we are interested in finding policies that improve on $\pi_0$ and avoid policies that perform worse than $\pi_0$. To replicate real world scenarios, we design an experiment where $\pi_0$ is a faulty policy ($\tau = 0.2$), that collects noisy ($\epsilon = 0.2$) interaction data, some of which is used to learn $\pi_{\theta^{\text{IPS}}}, \pi_{\theta^{\text{SN}}}$, and that we add to our discrete set of policies $\Pi_{k=4} = \{\pi_0, \pi^{\text{ideal}}, \pi_{\theta^{\text{IPS}}}, \pi_{\theta^{\text{SN}}}\}$. The splits for these experiments are the following: 70% of the data is used to create bandit feedback (20% is used to train $\pi_{\theta^{\text{IPS}}}, \pi_{\theta^{\text{SN}}}$ and 50% is used to evaluate policies based on estimators/upper bounds.) the rest is used to evaluate the true value of the policies. The goal is to measure the ability of our selection strategies to choose from $\Pi_{k=4}$, better performing policies than $\pi_0$. We thus define three possible outcomes: a strategy can select *worse* performing policies, *better* performing or the *best* policy. We compare selection strategies based on upper bounds to the commonly used estimators IPS and SNIPS. The hyperparameters of all bounds (the clipping parameter $M$ and $\lambda$) are set to $1/\sqrt{n}$. The comparison is conducted on the 11 datasets with 10 different seeds resulting in 110 scenarios. In addition to the plot in Figure 2, we collect the number of times each method selected the best policy ($\pi_*^{\text{S}}$), a better (**B**) or a worse (**W**) policy than $\pi_0$ for all datasets in Table 6. We can see that risk estimators can be unreliable, especially in small sample datasets, as they can choose worse performing policies than $\pi_0$, a catastrophic outcome in highly sensitive applications. Selecting policies based on upper bounds is more conservative, as it avoids completely poor performing policies. In addition, the tighter the bound, the better its percentage of time it selects the best policy: LS upper bound is less conservative and can find best policies more than any other bound, while never selecting poor performing policies.

Table 6: OPS: Number of times the worst, better or best policy was selected for each dataset.

| Dataset | IPS | | | SNIPS | | | SN-ES | | | cIPS-EB | | | IX | | | cIPS-L=1 | | | LS | | |
|---|---|---|---|---|---|---|---|---|---|---|---|---|---|---|---|---|---|---|---|---|---|
| | W | B | $\pi_*^s$ | W | B | $\pi_*^s$ | W | B | $\pi_*^s$ | W | B | $\pi_*^s$ | W | B | $\pi_*^s$ | W | B | $\pi_*^s$ | W | B | $\pi_*^s$ |
| **ecoli** | 2 | 6 | 2 | 4 | 1 | 5 | 0 | 10 | 0 | 0 | 10 | 0 | 0 | 7 | 3 | 0 | 10 | 0 | 0 | 6 | 4 |
| **arrhythmia** | 0 | 10 | 0 | 0 | 10 | 0 | 0 | 10 | 0 | 0 | 10 | 0 | 0 | 7 | 3 | 0 | 10 | 0 | 0 | 5 | 5 |
| **micro-mass** | 3 | 0 | 7 | 1 | 0 | 9 | 0 | 10 | 0 | 0 | 10 | 0 | 0 | 0 | 10 | 0 | 0 | 10 | 0 | 0 | 10 |
| **balance-scale** | 0 | 3 | 7 | 0 | 2 | 8 | 0 | 10 | 0 | 0 | 10 | 0 | 0 | 4 | 6 | 0 | 10 | 0 | 0 | 3 | 7 |
| **eating** | 3 | 2 | 5 | 2 | 1 | 7 | 0 | 10 | 0 | 0 | 10 | 0 | 0 | 4 | 6 | 0 | 8 | 2 | 0 | 4 | 6 |
| **vehicle** | 3 | 0 | 7 | 1 | 1 | 8 | 0 | 10 | 0 | 0 | 10 | 0 | 0 | 5 | 5 | 0 | 10 | 0 | 0 | 3 | 7 |
| **yeast** | 0 | 2 | 8 | 2 | 0 | 8 | 0 | 10 | 0 | 0 | 10 | 0 | 0 | 2 | 8 | 0 | 7 | 3 | 0 | 2 | 8 |
| **page-blocks** | 0 | 0 | 10 | 0 | 0 | 10 | 0 | 10 | 0 | 0 | 10 | 0 | 0 | 0 | 10 | 0 | 10 | 0 | 0 | 0 | 10 |
| **optdigits** | 0 | 1 | 9 | 0 | 0 | 10 | 0 | 10 | 0 | 0 | 10 | 0 | 0 | 1 | 9 | 0 | 3 | 7 | 0 | 1 | 9 |
| **satimage** | 0 | 0 | 10 | 0 | 0 | 10 | 0 | 10 | 0 | 0 | 10 | 0 | 0 | 0 | 10 | 0 | 7 | 3 | 0 | 0 | 10 |
| **kropt** | 0 | 0 | 10 | 0 | 0 | 10 | 0 | 10 | 0 | 0 | 0 | 10 | 0 | 0 | 10 | 0 | 0 | 10 | 0 | 0 | 10 |

## H.2 Off-policy learning

### H.2.1 Datasets

As described in the experiments section, we follow exactly the experimental design of Sakhi et al. [49], Aouali et al. [5] to conduct our PAC-Bayesian Off-Policy learning experiments. We however take the time to explain it in details. In this procedure, we need three splits: $D_l$ (of size $n_l$) to train the logging policy $\pi_0$, another split $D_c$ (of size $n_c$) to generate the logging feedback with $\pi_0$, and finally a test split $D_{test}$ (of size $n_{test}$) to compute the true risk $R(\pi)$ of any policy $\pi$. In our experiments, we split the training split $D_{train}$ (of size $N$) of the four datasets considered into $D_l$ ($n_l = 0.05N$) and $D_c$ ($n_c = 0.95N$) and use their test split $D_{test}$. The detailed statistics of the different splits can be found in Table 7. Recall that $K$ is the number of actions and $p$ the number of features.

Table 7: OPL: Detailed statistics of the splits used.

| Datasets | $N$ | $n_l$ | $n_c$ | $n_{test}$ | $K$ | $p$ |
|---|---|---|---|---|---|---|
| **MNIST** | 60 000 | 3000 | 57 000 | 10 000 | 10 | 784 |
| **FashionMNIST** | 60 000 | 3000 | 57 000 | 10 000 | 10 | 784 |
| **EMNIST-b** | 112 800 | 5640 | 107 160 | 18 800 | 47 | 784 |
| **NUS-WIDE-128** | 161 789 | 8089 | 153 700 | 107 859 | 81 | 128 |

### H.2.2 Policy class

In the PAC-Bayesian Learning paradigm, we are interested in the definition of policies as mixtures of decision rules:

$$\pi_Q(a|x) = \mathbb{E}_{f_\theta \sim Q}\left[\mathbb{1}\left[f_\theta(x) = a\right]\right], \qquad \forall (x,a) \in \mathcal{X} \times \mathcal{A}. \tag{56}$$

We use the Linear Gaussian Policy of Sakhi et al. [49]. To obtain these policies, we restrict $f_\theta$ to:

$$\forall x \in \mathcal{X}, \quad f_\theta(x) = \underset{a' \in \mathcal{A}}{\mathrm{argmax}}\left\{x^t \theta_{a'}\right\} \tag{57}$$

This results in a parameter $\theta$ of dimension $d = p \times K$ with $p$ the dimension of the features $\phi(x)$ and $K$ the number of actions. We also restrict the family of distributions $\mathcal{Q}_{d+1} = \{Q_{\boldsymbol{\mu},\sigma} = \mathcal{N}(\boldsymbol{\mu}, \sigma^2 I_d), \boldsymbol{\mu} \in \mathbb{R}^d, \sigma > 0\}$ to independent Gaussians with shared scale. Estimating the propensity of $a$ given $x$ reduces the computation to a one dimensional integral:

$$\pi_{\boldsymbol{\mu},\sigma}(a|x) = \mathbb{E}_{\epsilon \sim \mathcal{N}(0,1)}\left[\prod_{a' \neq a} \Phi\left(\epsilon + \frac{\phi(x)^T(\boldsymbol{\mu}_a - \boldsymbol{\mu}_{a'})}{\sigma||\phi(x)||}\right)\right]$$

with $\Phi$ the cumulative distribution function of the standard normal.

### H.2.3 Detailed hyperparameters

Contrary to previous work, our method does not require tuning any loss function hyperparameter over a hold out set. We do however need to choose parameters to optimize the policies.

**The logging policy** $\pi_0$. $\pi_0$ is trained on $D_l$ (supervised manner) with the following parameters: We use $L_2$ regularization of $10^{-4}$. This is used to prevent the logging policy $\pi_0$ from being close to deterministic, allowing efficient learning with importance sampling. We use Adam [30] with a learning rate of $10^{-1}$ for 10 epochs.

**Parameters of the bounds.** cIPS and cvcIPS: The clipping parameter $\tau$ is fixed to $1/K$ with $K$ the action size of the dataset and cvcIPS is used with $\xi = -0.5$ (the values used in Sakhi et al. [49]). ES: The exponential smoothing parameter $\alpha$ is fixed to $1 - 1/K$.

**Optimizing the bounds.** We use Adam [30] with a learning rate of $10^{-3}$ for 100 epochs. The gradient of **LIG** policies is a one dimensional integral, and is approximated using $S = 32$ samples.

$$\pi_{\boldsymbol{\mu},\sigma}(a|x) = \mathbb{E}_{\epsilon \sim \mathcal{N}(0,1)} \left[ \prod_{a' \neq a} \Phi \left( \epsilon + \frac{\phi(x)^T (\boldsymbol{\mu}_a - \boldsymbol{\mu}_{a'})}{\sigma ||\phi(x)||} \right) \right]$$

$$\approx \frac{1}{S} \sum_{s=1}^{S} \prod_{a' \neq a} \Phi \left( \epsilon_s + \frac{\phi(x)^T (\boldsymbol{\mu}_a - \boldsymbol{\mu}_{a'})}{\sigma ||\phi(x)||} \right) \quad \epsilon_1, ..., \epsilon_S \sim \mathcal{N}(0,1).$$

For all bounds, instead of fixing $\lambda$, we take a union bound over a discretized space of possible parameters $\Lambda$ of size $n_\Lambda = 100$ and for each iteration $j$ of the optimization procedure, we take $\lambda_j \in \Lambda$ that minimizes the estimated bound and proceed to compute the gradient w.r.t $\mu$ and $\sigma$ with $\lambda_j$.

### H.2.4 Detailed results

In addition to the results of Table 2, we also provide a more detailed view of the results here. For each $\alpha$ and dataset, we average both $\{\mathcal{GR}, R\}$ over the 10 seeds and plot them in Figure 6 and Figure 5. Note that the error bars are too small $\sigma/\sqrt{10} \approx 0.001$ and all our results in these graphs are significant. We observe that the LS PAC-Bayesian bound improves substantially on its competitors in terms of the guaranteed risk, especially on MNIST and FashionMNIST and also obtains the best performing policies, on par with the IX bound in the majority of scenarios.

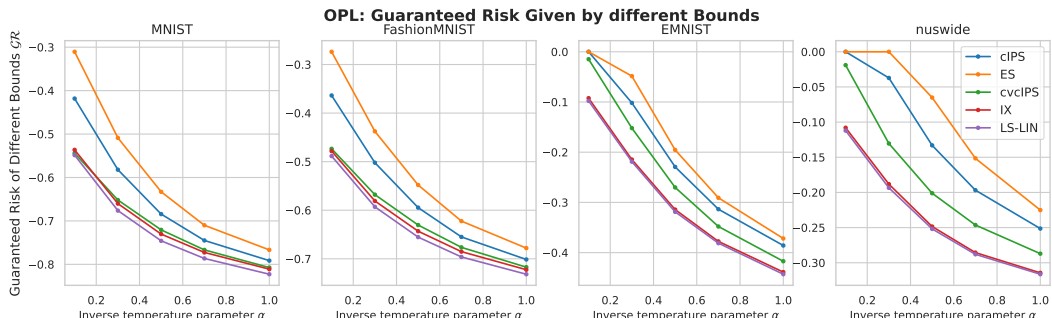

Figure 5: OPL: Guaranteed Risk given by the different bounds. We observe that our `LS-LIN` dominates all other bounds. `IX` comes close, especially on EMNIST and nuswide

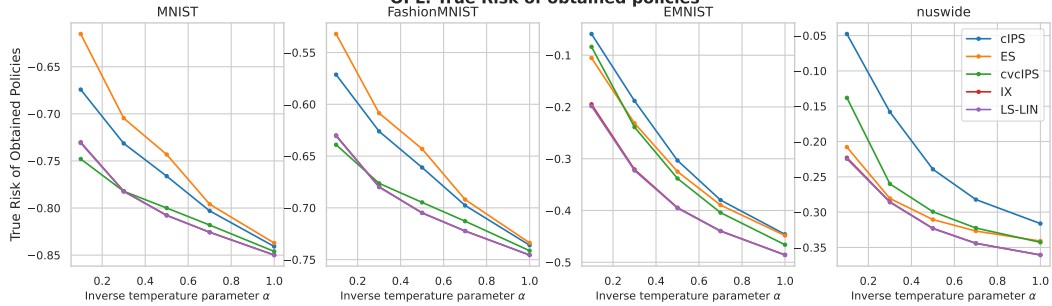

Figure 6: OPL: True risk of obtained policies after minimizing the PAC-Bayesian bounds. We observe that `LS-LIN` and `IX` are hardly distinguishable, they both give the best policies in the majority of scenarios.

