# OpenReview forum: "Logarithmic Smoothing for Pessimistic Off-Policy Evaluation, Selection and Learning"
_NeurIPS.cc/2024/Conference — NeurIPS 2024 spotlight_

### Official Review · Reviewer_j8sh · 2024-06-28

**Soundness:** 3
**Presentation:** 3
**Contribution:** 3
**Rating:** 7
**Confidence:** 2

**Summary:**

The paper considers the offline contextual bandit problem. The authors consider a class of reward estimators for this setting that is a regularization of Inverse Propensity Scoring (IPS - aka importance sampling). A general concentration result is provided for this class of estimators. This is used to provide a tight result for an existing clipping IPS estimator and to construct a new Logarithmic Smoothing (LS) estimator. The resulting estimator is pessimistic by design, making it immediately applicable to the offline contextual bandit problem.
The authors use it to derive bounds for policy evaluation and selection and also for policy learning in the Bayesian setting. Experimental results also support the usefulness of the estimator.

**Strengths:**

I am only broadly familiar with this line of research making it hard for me to properly contextualize its contributions.

1. The proposed estimator is novel and has nice properties.
2. As the name suggests, the estimator is smooth making it potentially easy to optimize.
3. The application to contextual bandits is interesting.
4. The experimental results are positive.
5. The overall writing is good and clear.

**Weaknesses:**

1. A more explicit comparison with existing concentration\contextual bandit bounds is missing. The authors explain that their bound is better but this is somewhat vague, especially if the reader is not already an expert in this field.

2. In line 155 the authors explain that their result can be derived from [1, Lemma 1.3]. Does this mean that the LS estimator has previously been suggested or only that an alternative proof technique exists for its concentration bound?

3. Performance seems very close to that of IX

4. The main body of the paper does not include any explanation of the techniques used. This can be a proof sketch for the concentration bound or a discussion comparing your approach to existing techniques. Can you provide such an explanation in your response?

5. The notation U(pi) appears without definition in line 281. I assume it's defined in one of the references but should also be defined in this paper for completeness. (Please include an explanation in your response)

Typo:
line 98: one of the brackets is reversed in the definition of h

**Questions:**

See above.

---

> ### Author Rebuttal · Authors · 2024-08-05
>
> First, we would like to thank you very much for your positive review acknowledging the quality of our work. We hope our response addresses your questions and increases your confidence in our work. We will consider the points raised when updating the manuscript.
>
> **(1) Bound Comparisons**
>
> Given the 9-page limit for the submitted paper, the tightness claim is thoroughly defended in the Appendix. We directed the reader to the full, explicit comparisons and discussions in the Appendix whenever a statement is made. Specifically:
>
> - **Appendix E.2 for OPE**:
>   - Comparison of the global clipping bound ($U^\lambda_1$) with the empirical Bernstein bound for clipping (Appendix E.2.1).
>   - Comparison of the LS bound ($U^\lambda_\infty$) with the IX bound (Appendix E.2.2).
>
> - **Appendix E.4 for OPL**:
>   - Comparison of the PAC-Bayesian LS bound with the conditional Bernstein bound, the exponential smoothing bound, and the IX bound (Appendix E.4).
>
> All these comparisons show that our bounds are tighter.
>
> To respect the 9-page limit, the paper focuses on presenting the methodology, introducing the new estimator, and proving the versatility of the approach. With the additional page available in the camera-ready version, we will move these comparisons to the main text.
>
> **(2) Alternative Proof**
>
> This is the first time the LS estimator has been proposed, with [1, Lemma 1.3] only providing an alternative proof of its concentration bound. We chose our proof technique for its generality, which is appealing for two reasons. First, it yields a family of empirical upper bounds for a wide range of regularized IPS estimators, such as clipping. Second, within this family, the LS estimator emerged as the one achieving the tightest bound. This strongly motivates the LS estimator, in contrast to defining it in advance (without a clear rationale for its selection) and applying [1, Lemma 1.3] for its concentration.
>
> **(3) Performance Close to IX**:
>
> Our experiments demonstrate the superiority of LS compared to IX in OPE and OPS. Even in OPL, LS has a better guaranteed risk than IX, and they are only comparable in terms of the risk of the learned policy. Morevoer, a formal comparison of the LS bound with IX is provided in Appendix E.2.2, showing that LS consistently outperforms IX (for any $\lambda$, in any scenario). It also highlights that the LS bound significantly outperforms the IX bound when the target policy $\pi$ is not deterministic and/or in the low data regime (starting from line 746). This is further confirmed by the detailed results in Appendix H.1.2.
>
> **(4) Proof Technique Used**
>
> In the main text (line 121), we state that we use the Chernoff bound with a careful analysis of the moment generating function. Specifically, we prove the monotonicity of the logarithm's residual (the difference between the logarithm function and its Taylor expansion) to derive our results. Our approach differs from existing bounds by using the residual function in combination with the Chernoff bound.
>
> **(5) The Notation $U(\pi)$ and Typo**
>
> Thank you for pointing this out. $U(\pi)$ refers to a generic bound evaluated for the target policy $\pi$. We will include its explanation in the updated manuscript. We also thank you for highlighting the typo.

---

> > ### Comment · Reviewer_j8sh · 2024-08-13
> > **Response**
> >
> > Thank you for responding to my concerns, I will keep my score.
> >
> > A small side note: the abbreviation LS is usually associated with Least Squares. If there is another sensible naming option it will probably help avoid confusing the two :)

---

### Official Review · Reviewer_PPYn · 2024-07-12

**Soundness:** 2
**Presentation:** 1
**Contribution:** 2
**Rating:** 3
**Confidence:** 5

**Summary:**

The authors propose empirical concentration inequalities for off-policy evaluation that apply to several forms of (smoothed) IPS, which are claimed to be tighter than the results in existing works. These bounds are then used to derive policy learning guarantees that inherit the properties of the concentration inequalities.

**Strengths:**

I appreciate that the authors have applied their method to OPE, OPS, OPL, and also provided some experiments.

I did not read the appendix nor check the correctness of the analysis in detail, but from a quick glance it appears that the authors were careful to provide rigorous and well-organized proofs.

**Weaknesses:**

My biggest criticism is that the authors have not justified *in the main body* their claim that "LS is provably tighter than its competitors" (L12) for any of the results, including the concentration inequalities (Prop 1, Cor 3, Cor 4) and the policy learning guarantee (e.g., Prop 6).

Since these claims are the whole premise for the paper, their justification should be a central pursuit and only stating "x is in Appendix y" (L147, 178, 195) is hugely insufficient.

For example, I would have liked to see a discussion on (possibly even in graphs):
- For the choices of $h$ described in (4), when do the bounds in Prop 1, Cor 3, and Cor 4 improve over the bounds from their respective papers?
- Is $h*$ (the tightest choice) better than all of the above?
- Does this hold for all hyperparameter choices, e.g. $\lambda$ and $L$?
- How does the computational complexity of calculating the bounds in Prop 1, Cor 3, Cor 4 hold up relative to their competitors?
- Exactly how does this lead to downstream policy learning improvements?

Lastly, I found the overall technical presentation to be relatively poor, and I'll give a few examples:
- The condition (C1) from Section 2 ("Regularized IPS") that all results depend on is never explicitly defined, and it should be an assumption that is called in every proceeding proposition/theorem statement.
- Shouldn't (11) be framed in, e.g., a lemma environment?
- The term "pessimism" is overloaded, e.g., for "high-probability upper bounds" in L111 but also for an in-expectation variant in Eq. (5), which is slightly unusual (and I'm pretty sure not the way it's used in [26]) but not recalled again in the main body so I'm not sure what it's for (perhaps the proof of Prop 1).

**Questions:**

In addition to the ones in "Weaknesses," I have a specific question about Proposition 6. The gold standard in offline policy selection is a bound in the form of $R(\pi) - R(\widehat\pi) \le \lambda S(\pi) + \varepsilon$ for any comparator policy $\pi$ rather than the optimal one $\pi^*$ (see [26] and [Wang 2024] and [Xie 2021]). The former is strictly more general -- can you write your bound in such a form?


**References**

Wang, L., Krishnamurthy, A., & Slivkins, A. (2024, April). Oracle-efficient pessimism: Offline policy optimization in contextual bandits. In International Conference on Artificial Intelligence and Statistics (pp. 766-774). PMLR.

Xie, T., Cheng, C. A., Jiang, N., Mineiro, P., & Agarwal, A. (2021). Bellman-consistent pessimism for offline reinforcement learning. Advances in neural information processing systems, 34, 6683-6694.

**Limitations:**

I do not believe the authors have fully discussed the limitations of their method (see "Weaknesses").

---

> ### Author Rebuttal · Authors · 2024-08-05
>
> First of all, we would like to thank you for your review and we hope that our response answers your questions and clears out misunderstandings. We think that your comments can be completely addressed and we hope this will lead you to increase your score.
>
> **Answer to the main criticism**
>
> First, we want to clarify that our claim "LS is provably tighter than its competitors" is theoretically proved by comparing (Eq. (13), Corollary 4) to existing bounds. This discussion was included in the Appendix and will be moved to the main text in the revised version as we will have additional space. This claim is also supported empirically by plotting the bounds (see answer below).
>
> Second, there are many other equally important contributions central to the paper. We derive novel, high-probability empirical upper bounds applicable to a large family of "regularized IPS". These bounds are analyzed, compared, and minimized to obtain a new estimator, Logarithmic Smoothing (LS), which has favorable properties. Its empirical bound (Eq. (13), Corollary 4) is tighter than those of competitors, and the estimator is pessimistic by design, showing excellent performance in extensive OPE, OPS, and OPL experiments.
>
> **Answers to the points raised**
>
> **1-** For the choices of described in (4), when do the bounds in Prop 1, Cor 3, and Cor 4 improve over the bounds from their respective papers?
>
> First, to clear up any misunderstanding, we did not claim that our bounds are better than existing ones for any choice of $h$. We claimed that the LS bound (Eq. (13): Corollary 4, evaluated at its minimizer) is tighter than the existing bounds derived for their respective estimators.
>
> Now to address this question, we focus on the choices in (4): Clipping, Exponential Smoothing (ES), and Implicit Exploration (IX). Harmonic and Shrinkage do not come with empirical upper bounds, so they are omitted from this comparison. From the paper, we __theoretically__ prove:
>
> - Our bound with $L = 1$ (Cor 3) with $\lambda \sim \mathcal{O}(1/\sqrt{n})$ applied to Clipping is tighter than the empirical Bernstein bound for Clipping [56], especially when $\pi$ is different from $\pi_0$ (Appendix E.2.1).
> - ES only provides a PAC-Bayesian bound, which is proven to be worse than ours when applied to ES, for any $\lambda$ (see Appendix E.4, starting line 818).
> - The LS bound $L = \infty$ (Cor 4), evaluated at $h_{*, \infty}$ (Eq. (13)), is tighter than the IX bound for any $\lambda$ (Appendix E.2.2).
>
>
> **2-** Is $h^*$ better than all of the above?
>
> __Theoretically__, we can prove:
>
> - The bound of LS ($h_{*, \infty}$) with any $\lambda$ is tighter than the IX bound (Appendix E.2.2).
> - The bound of Global Clipping ($h_{*, 1}$) with $\lambda \sim \mathcal{O}(1/\sqrt{n})$ is tighter than the empirical Bernstein bound (Appendix E.2.1).
> - The bound of LS ($h_{*, \infty}$) with any $\lambda$ is tighter than the bound of Global Clipping (Proposition 5).
> - Thus, the LS bound is better than all known evaluation bounds: LS is better than IX and Global Clipping for any $\lambda$. Since Global Clipping is tighter than empirical Bernstein when $\lambda \sim \mathcal{O}(1/\sqrt{n})$, LS is also better than empirical Bernstein for these values of $\lambda$.
> - For learning, the PAC-Bayesian bound of LS ($h_{*, \infty}$) is tighter than all known PAC-Bayesian bounds, for any $\lambda$ (Appendix E.4).
>
>
> __Additional plots.__ The claims in __1-__ and __2-__ above are supported __empirically__ (see global response). These plots support even stronger claims such as showing that LS bound is better than empirical Bernstein for any $\lambda$. In these plots, we compare Proposition 1 with different $L$ for different choices of $h$. We also compare our bounds (Global Clipping and LS) to Proposition 1 (for different $L$) evaluated in Clipping and IX and to their respective bounds (empirical Bernstein and the IX bound). Finally, a comparison of the bounds for different values of $\lambda$ is provided. In particular, these plots also address your third point __3-__ Does this hold for all hyperparameter choices, e.g.  $L$ and $\lambda$?
>
> **4-** How does the computational complexity of calculating the bounds in Prop 1, Cor 3, Cor 4 hold up relative to their competitors?
>
> Our bounds are tractable, in contrast to many existing studies that provide intractable bounds (e.g., [37, 56]). Tractable bounds do exist in the literature, with the IX bounds being the most computationally efficient. Cor 4 has a similar complexity to the IX bound, as it only requires computing the LS estimator without any additional high-order terms.
>
> **5-** Exactly how does this lead to downstream policy learning improvements?
>
> Tighter bounds lead to policy selection and learning strategies with provably better regret/suboptimality. For example, the suboptimality derived for the LS estimator is tighter than that of the IX estimator in all scenarios (Appendix E.3 for OPS and Appendix E.5 for OPL). This typically translates into better empirical results, as supported by our experiments.
>
> **The technical presentation**
>
> - Condition **(C1)** is explicitly defined in line 100. We will refer to this condition in our propositions.
> - Eq. (11) is the derived minimiser of Cor 3 and was not framed as a proposition to ease reading.
> - The in-expectation pessimism is indeed used to prove Proposition 1, and we will drop its naming to offload the term 'pessimism'.
>
> **Subotimality with a comparator** $\pi$ **instead of** $\pi^*$
>
> The technique used to derive the suboptimality does not rely on any specific property of $\pi^*$ and can be directly applied to prove the bound for any $\pi$ by replacing $\pi^*$ with $\pi$. We will add this more general result to the Appendix. We aim to identify the optimal policy $\pi^*$, which is why the suboptimalities are expressed w.r.t. $\pi^*$. This is common in the literature [26, 27].

---

> > ### Comment · Reviewer_PPYn · 2024-08-14
> >
> > Thank you for your detailed response. It's plausible that the bounds in the paper are interesting contributions and may be tighter than existing ones. However, the submission in its current form does not make this argument convincingly. As it is the central tenet of the paper, I believe that all of the discussions, results, and comparisons above should be a central focus of the main body.
> >
> > Given the amount of content that I feel is omitted and worth analyzing more deeply, I do not feel that one round of revision is sufficient to address my concerns, and I stand by my original review.

---

> > > ### Author Response · Authors · 2024-08-14
> > >
> > > Thank you for your feedback. However, we strongly disagree with the reviewer's assessment. The comment that our bounds "may be" tighter than existing ones and that the paper "does not make this argument convincingly" seems to overlook the rigorous evidence we provided. We demonstrated the tightness of the LS bound (and our other bounds) through both theoretical analysis (in the paper) and empirical validation (in the OPE experiments and in the rebuttal), going beyond the typical approach of relying solely on one or the other.
> > >
> > > We would like to clarify that our tightness claims are made in the main text, with proofs provided in the Appendix. Redirecting readers to proofs in the Appendix is standard in the field, and we have explicitly stated our intention to include them along with additional discussions in the camera-ready version. This can be easily accommodated with the additional page, as other reviewers did not request major revisions.
> > >
> > > Finally, it would have been helpful if the reviewer had specified which concerns remain unaddressed by our rebuttal. The statement "I do not feel that one round of revision is sufficient to address my concerns" lacks explanation, leaving us unsure of which parts of the rebuttal did not address the reviewer's concerns, as we aimed to thoroughly address all points raised in the initial review.

---

### Official Review · Reviewer_J96t · 2024-07-13

**Soundness:** 3
**Presentation:** 2
**Contribution:** 3
**Rating:** 6
**Confidence:** 3

**Summary:**

This paper studies log-algorithmic smoothing of importance weight for off-policy learning. The proposed smoothing technique can be seen as a differentiable variant of clipping, which is useful for variance reduction for OPL. The paper also analyzes the PAC-Bayes learning bound of the proposed OPL method, characterized by the KL divergence with the logging policy, showing that the proposed method achieves a tighter bound than baselines, including simple clipping. The experiment also shows that the proposed method has tighter bounds than baselines and enables more accurate off-policy selection.

**Strengths:**

- **Reasonable formulation based on theoretical analysis**: The proposed method is derived from a tight upper bound of the policy's risk. Also, the proposed method has an interpretation as soft, differentiable clipping. The technique is well-motivated and is reasonable to interpret.

- **PAC-Bayes learning bound**: A sub-optimality form is derived, and it is also easy to interpret as a pessimistic approach, which should be acknowledged.

- **Experiments on various tasks**: The paper evaluates the proposed approach in upper bound derivation, off-policy selection, and off-policy learning. The experiment results show the wide applicability of the proposed method in many OPE/OPL-related tasks.

**Weaknesses:**

- **Connection to Metelli et al. 2021 is not clear**: Metelli et al. 2021 also considers the importance of weight differential and shows that the proposed method achieves a Subgaussian rate. Similar to the reviewed paper, Metelli et al. 2021 also have a KL divergence term in the theoretical analysis. While the proposed method adequately differs from Metelli et al. 2021, and the paper does cite it, the paper does not mention Metelli et al. 2021 in the related work in detail. Since the motivation and contributions are similar, a detailed discussion on the advantages and the differences would be appreciated.

- **Baselines in the experiments**: As mentioned above, Metelli et al. 2021 propose a similar idea that can be used as a baseline in experiments. Comparing with advanced regularization techniques such as shrinkage (Su et al. 2020) would also be informative.

(Metelli et al. 2021) Subgaussian and Differentiable Importance Sampling for Off-Policy Evaluation and Learning. Alberto Maria Metelli, Alessio Russo, Marcello Restelli. NeurIPS, 2021.

(Su et al. 2020) Doubly robust off-policy evaluation with shrinkage. Yi Su, Maria Dimakopoulou, Akshay Krishnamurthy, Miroslav Dudík. ICML, 2020.

**Questions:**

- What are the connections with Metelli et al. 2021? (See weaknesses for the detailed comments.)

- How does OPL work with the varying performance of the behavior policy? In my understanding, the policy will be pessimistic in out-of-distribution, but seeing how it works in experiments would be informative for readers.

**Limitations:**

Missing connection with a similar idea. See the weaknesses for the details.

---

> ### Author Rebuttal · Authors · 2024-08-05
>
> First of all, we would like to thank you for your positive review, and we hope that our response addresses your questions and clears up any misunderstandings.
>
> **(1) Connection to Metelli et al. 2021**
>
> Our Logarithmic Smoothing (LS) estimator and the Harmonic estimator of Metelli et al. (2021) share similarities in how they avoid hard clipping on importance weights while obtaining sub-Gaussian concentration. However, they differ in several key aspects, which we list below:
> - **The motivation:** Metelli et al. (2021) introduced the Harmonic estimator specifically to mitigate the heavy tail of IPS and focused their study on it. In contrast, LS is motivated differently. We begin by deriving a high-probability upper bound that applies to a large family of regularized IPS estimators, including the Harmonic estimator. We then identify the estimator within this family that achieves the tightest possible bound. This turns out to be LS, a novel estimator that logarithmically smooths the importance weights. While the Harmonic estimator can be used with our bounds, the LS estimator provides tighter results.
> - **No empirical upper bound for the Harmonic estimator:** Metelli et al. (2021) study the properties of the Harmonic estimator and derive a concentration inequality (Theorem 5.1) that, in the case of contextual bandits, involves the divergence
> $$I_{\alpha}(\pi, \pi_0) = \mathbb{E}_{x \sim \nu, a \sim \pi_0(\cdot|x)} \left[  \left(\frac{\pi(a|x)}{\pi_0(a|x)} \right)^\alpha \right],$$
> between the target policy $\pi$ and the behavior policy $\pi_0$. This quantity makes the upper bound **intractable** because it requires computing an expectation under the unknown distribution of contexts $\nu$. Consequently, the derived concentration provides insight into the estimator's behavior but cannot be used to implement **pessimism** as it is not empirical. This contrasts with our fully empirical bounds that can be used directly for efficient pessimism.
> - **The LS estimator enjoys a better concentration:** Equation (2) (Page 5) of Metelli et al. (2021) proves the sub-Gaussian concentration of the Harmonic estimator. We also prove the sub-Gaussian property of the LS estimator in Proposition 13 in Appendix E.1 and briefly compare it to Metelli et al. (2021). Our findings show that the LS estimator enjoys a better sub-Gaussianity constant than the Harmonic estimator.
> - **Metelli et. al. 2021 do not have a KL term:** Metelli et al. (2021) only provide the concentration property of the Harmonic estimator and did not analyze the estimator in off-policy learning. Their concentration uses $I_{\alpha}(\pi, \pi_0)$, which simplifies to the theoretical second moment of the importance weights when $\alpha = 2$. This differs from the KL divergence between the distributions inducing the policies, which is proper to the PAC-Bayesian analysis of off-policy learning [5, 20, 48].
>
> **(2) Baselines in the experiments**
>
> In our experiments, we aim to compare the ability of estimators to implement efficient **pessimism**. Therefore, we only include estimators from the literature that come with **empirical/tractable** upper bounds. This is why we include clipped IPS, SNIPS, and IX, but not Harmonic and Shrinkage, as these do not provide **empirical/tractable** upper bounds for pessimism.
>
> **(3) How does OPL work with the varying performance of the behavior policy?**
>
> These experiments are given in Appendix H.2.4. Specifically, we conducted OPL experiments while varying the inverse temperature $\alpha$ of the behavior policy. Changing $\alpha$ interpolates between a uniform policy and a good behavior policy, resulting in different performance levels of the behavior policy. The main body of the paper presents aggregated results for ease of exposition, while detailed performance for each value of $\alpha$ can be found in Appendix H.2.4. Generally, the performance of the learned policy for all methods decreases as the behavior policy's performance decreases. Overall, the performance gap between LS and the baselines widens as the behavior policy's performance gets lower.

---

> > ### Comment · Reviewer_J96t · 2024-08-10
> >
> > Thank you for providing responses to the questions. After reading the rebuttals, the difference with the Harmonic estimator (especially the upper bound analysis) became clearer. I believe this paper is worth sharing with the community, and I would keep my initial evaluation.

---

### Official Review · Reviewer_VVmk · 2024-07-15

**Soundness:** 4
**Presentation:** 4
**Contribution:** 4
**Rating:** 10
**Confidence:** 5

**Summary:**

Policy evaluation, selection and optimization are considered in the context of offline contextual bandits, where i.i.d. data with a known behavior policy is given. The authors set out to study a generalization of importance weighted policy evaluation; for this they start from a general formulation that computes a value for all data observations, which are then averaged. The free "parameter" here is $h$, the function that assigns a value given an observation (of a context, associated action, and cost). A tight, general, high probability upper bound on the expected cost of a fixed target policy is derived first. Specific choices for the map $h$ are then derived based on minimizing this upper bound. Two practical solutions to this optimization problem are studied in more details: Global clipping and "logarithmic smoothing". Results are then derived for both policy selection and optimization.

**Strengths:**

Novel ideas, novel results, good empirical results.

**Weaknesses:**

Despite saying  that the methodology of paper [31] is adopted, this is only partially done. Why deviate from the evaluation in [31]? I expected an explanation of this.

**Questions:**

Can you explain why you did not follow the protocol and reported values of [31]?

---

> ### Author Rebuttal · Authors · 2024-08-05
>
> First of all, we would like to thank you for your positive feedback acknowledging the quality of our work, and we hope that our response addresses your question.
>
> In our paper, we adopt the experimental design of [31] for both pessimistic policy evaluation and selection. The authors of [31] identified technical errors in their derivations after the publication of the paper, which they corrected by making stronger assumptions (one-hot rewards) and changing their proof technique. These changes resulted in loosened bounds and experimental results that were inconsistent with their initial findings. For more details, please refer to Appendix A of the updated arXiv version of [31].
>
> These modifications are also reflected in the official GitHub repository of [31], which we used for part of our experiments. Due to these discrepancies, we reran all experiments from scratch for better reproducibility. Our code is included in the supplementary material and can be run to reproduce our results.
>
>
>
> [31] Ilja Kuzborskij, Claire Vernade, Andras Gyorgy, and Csaba Szepesvári. Confident off-policy evaluation and selection through self-normalized importance weighting. In International Conference on Artificial Intelligence and Statistics, pages 640–648. PMLR, 2021.

---

### Author Rebuttal · Authors · 2024-08-06

We are very grateful to the reviewers and AC for their valuable time. We attach to our rebuttal additional plots comparing the properties of the bounds for three different datasets and supporting empirically our theoretical findings, showing that the LS bound is tighter than its competitors.

**Figure 1- Comparison of Proposition 1 for different IPS regularisation functions $h$ and different $L$.**

For this plot, $\lambda$ is fixed to $1/\sqrt{n}$ while we vary the regularization function $h$ and the moment order $L$. We observe that the performance of the bound differs depending on the choice of $h$ and $L$. Additionally, the optimality of a regularization function $h$ depends on the value of $L$. For example, on the balance-scale dataset where the difference between methods is more apparent, Clipping (orange curve) is better than IPS without any regularization (blue curve) for finite values of $L$, while IPS becomes better when $L \to \infty$. This is why, in our paper, we seek the minimizer $h$ for our bounds to obtain the tightest results specific to values ($L = 1$ and $L \to \infty$), finding theoretically that the minimizer for $L=1$ differs from that for $L \to \infty$. These empirical bound plots confirm this aspect of our theory.

**Figure 2- Comparison with Clipping and its empirical Bernstein bound.**

For this plot, $\lambda$ is fixed to $1/\sqrt{n}$ and $M = \sqrt{n}$ (where $M$ is the hyper-parameter of Clipping as shown in Eq. (4)). We observe that Proposition 1 applied to Clipping, as well as our other bounds (Global Clipping and LS), outperform Clipping and its empirical Bernstein bound in all scenarios.

**Figure 3- Comparison with IX and its bound.**

For this plot, $\lambda$ is fixed to $1/\sqrt{n}$. We observe that our Proposition 1 evaluated in IX does not perform well compared to the specialized IX bound. Global Clipping is comparable to the IX bound and is tighter in scenarios with sufficient data (2nd and 3rd plot). Finally, our LS bound is always tighter than the IX bound, as proven theoretically in our paper (Appendix E.2.2).

**Figure 4- Comparison of the bounds for different values of $\lambda$.**

For this plot, we set $M = 1/\lambda$ for the empirical Bernstein bound of Clipping and vary $\lambda$ for all our bounds. As already proven in our paper, the LS bound is better than both Global Clipping (Proposition 5) and the IX bound (Appendix E.2.2) in all scenarios, for all values of $\lambda$. We also observe that all our bounds outperform the empirical Bernstein bound of Clipping, especially in the region of $\lambda \sim \mathcal{O}(1/\sqrt{n})$, as shown theoretically in Appendix E.2.1.

---

### Decision · Program_Chairs · 2024-09-25

**Decision:**

Accept (spotlight)

**Comment:**

This paper gives strong results for off-policy policy evaluation using importance sampling, where the importance ratios can be transformed using a class of functions to navigate the bias-variance tradeoff. The formulation makes the result quite general and the authors give a variety of examples for typically used transformation functions which fit their conditions, as well as use the resulting bounds to identify near-optimal choices. They further study the impacts of these ideas for policy learning, and evaluate their results empirically.
The results presented here are very nice, but the presentation could use a fair bit of improvement. This is reflected in several reviewers having difficulty understanding parts of the paper. Particularly the writing around the logarithmic smoothing and its optimality needs to be improved so that it is very clear that the optimality claim is rigorous.
There are questions about the empirical evaluation as well, and I encourage the authors to address any discrepancies between the setup of [31] and their work.